# Functional and molecular characterization of a non-human primate model of autism spectrum disorder shows similarity with the human disease

Satoshi Watanabe [1✉], Tohru Kurotani[1], Tomofumi Oga [1], Jun Noguchi[1], Risa Isoda[1], Akiko Nakagami[1,2], Kazuhisa Sakai[1], Keiko Nakagaki[1], Kayo Sumida[3], Kohei Hoshino [4], Koichi Saito[3], Izuru Miyawaki[4], Masayuki Sekiguchi[5], Keiji Wada[5], Takafumi Minamimoto [6] & Noritaka Ichinohe[1✉]

Autism spectrum disorder (ASD) is a multifactorial disorder with characteristic synaptic and gene expression changes. Early intervention during childhood is thought to benefit prognosis. Here, we examined the changes in cortical synaptogenesis, synaptic function, and gene expression from birth to the juvenile stage in a marmoset model of ASD induced by valproic acid (VPA) treatment. Early postnatally, synaptogenesis was reduced in this model, while juvenile-age VPA-treated marmosets showed increased synaptogenesis, similar to observations in human tissue. During infancy, synaptic plasticity transiently increased and was associated with altered vocalization. Synaptogenesis-related genes were downregulated early postnatally. At three months of age, the differentially expressed genes were associated with circuit remodeling, similar to the expression changes observed in humans. In summary, we provide a functional and molecular characterization of a non-human primate model of ASD, highlighting its similarity to features observed in human ASD.

[1] Department of Ultrastructural Research, National Institute of Neuroscience, National Center of Neurology and Psychiatry, Kodaira, Tokyo, Japan.
[2] Department of Psychology, Japan Women's University, Kawasaki, Kanagawa, Japan. [3] Environmental Health Science Laboratory, Sumitomo Chemical Co., Ltd., Konohana-ku, Osaka, Japan. [4] Preclinical Research Laboratories, Sumitomo Dainippon Pharma Co., Ltd., Konohana-ku, Osaka, Japan. [5] Department of Degenerative Neurological Diseases, National Institute of Neuroscience, National Center of Neurology and Psychiatry, Kodaira, Tokyo, Japan. [6] Department of Functional Brain Imaging, National Institutes for Quantum and Radiological Science and Technology, Chiba, Chiba, Japan. ✉email: s-watanabe@ncnp.go.jp; nichino@ncnp.go.jp

Autism spectrum disorder (ASD) is a highly prevalent neurodevelopmental disorder that affects >1% of the population, of which 85% are idiopathic with no direct evidence of specific genetic causes[1,2]. ASD is characterized by impaired social interaction and communication, repetitive behaviors, and restricted interests. Currently, its core symptoms cannot be cured and there is a need to develop pharmacological therapy. People with ASD typically show delayed development of social and language skills and are diagnosed at ~3 years old. However, early behavioral signs are predictive of pre-existing abnormalities in circuit development. Early behavioral intervention is recommended for children diagnosed with ASD, and better outcomes are expected if it is started earlier[3]. Thus, there is an urgent need to identify biological abnormalities both to understand the pathogenesis and to develop effective pharmacological treatments that can be started during the early developmental stage.

ASD is often discussed as a synaptopathy and there is evidence regarding synaptic dysfunction at both the structural and molecular levels. Dendritic spines in cortical neurons, which are postsynaptic to excitatory synapses and a proxy for synaptic density, are affected in people with ASD[4,5]. Consistent with this, multiple ASD-related genes are involved in synaptic functions[6]. Normal synaptic development proceeds in a precisely regulated manner, where excessive synaptic formation in the early postnatal stage is followed by pruning[7–9]. Synaptic development involves two processes: genetically programmed synaptogenesis and activity-dependent remodeling. Impairment of these processes is likely to adversely affect normal circuit generation with consequent ASD symptoms.

ASD affects primate-specific brain areas involved in social functions unique to primates (e.g., the medial prefrontal cortex)[10,11]. Moreover, the developmental expression pattern of some genes is primate specific[12,13], and proteome data suggest primate-specific synaptic functions[14]. Different molecular networks between humans and rodents may limit the utility of rodent models for human diseases, and therefore, ASD primate models have advantages over rodent models (also see ref. [15]). The common marmoset (Callithrix jacchus), a small New World monkey, is well suited for studies on neurodevelopmental diseases, and previous studies from our laboratory have described similarities in neuronal structure and synaptic development to humans[8,16]. Further advantages are that marmosets are easy to handle, mature rapidly, and are reproductively efficient. We established an ASD marmoset model by administering valproic acid (VPA) in utero[17,18]. VPA, an antiepileptic drug, increases the risk of ASD in offspring[19] by inhibiting histone deacetylase (HDAC) and DNA methylation[20]; furthermore, it has been used to generate ASD model animals[21,22]. VPA-exposed marmoset offspring present with abnormalities in social behavior[17,18], axon bundle structure[23], and neuroimmune cells[24], which are consistent with human data.

During early postnatal development, the brain experiences drastic changes in neural circuit organization and gene expression. Synaptogenesis is highest at the neonatal period[7], and experience-dependent remodeling of the circuit follows in childhood[25]. Indeed, sensitive periods of language acquisition and social learning occur in childhood[26]. Gene expression also changes rapidly during the neonatal period as synapses develop[12,27]. Genes with similar developmental dynamics form a module of functionally related genes[27]. Temporally related modulations in synaptic structure or function and gene expression will link ASD-related synaptic phenotypes with molecular functions. Accordingly, we investigated synaptic development using acute brain slices from the dorsomedial prefrontal cortex of a VPA-induced ASD marmoset model at the ages of birth (0 M),

3 months (3 M), and 6 months (6 M), which can be taken to correspond to neonatal, infancy, and puberty periods, respectively[8,16]. We also analyzed gene expression modulations at these ages. The present study indicated concurrent changes in synaptic and molecular phenotypes with development. Synapses of the model animal were underdeveloped in neonates, which was different from synaptic overdevelopment commonly observed in the model at puberty and in human ASD. Synaptic plasticity was transiently enhanced during infancy. We also clarified behavioral outcomes at infancy, since ASD symptoms appear in childhood. Clustering of genes based on the temporal profile of modulation revealed genes associated with synaptogenesis and synaptic remodeling. Molecular phenotypes of human idiopathic ASD were well replicated in this model after infancy in a large number of coexpression modules but only partially in various rodent models, suggesting a closer molecular pathway in the primate model that is affected in ASD. This study demonstrates the importance of early phenotypes in the pathogenesis of ASD and suggests novel molecular targets for early-age intervention and good outcomes throughout life.

## Results

**Early underdevelopment and late overdevelopment of synapses in VPA-exposed marmosets.** Among humans, there is a rapid increase in dendritic spines in cortical neurons until childhood. The spine density in the prefrontal cortex peaks at ~2–3 years of age and is followed by a decrease through synaptic pruning[7,28]. The age of 3 years is the typical diagnosis age for ASD, with symptoms becoming evident. Studies on the postmortem brain from people with ASD have reported a higher density of dendritic spines compared to typically developed people after childhood, either due to exaggerated spine formation or incomplete pruning through synaptic development[4,29]. Dendritic spines in the prefrontal cortex of typically developed marmosets have shown a similar trajectory of development with a peak spine density at 3 M and a subsequent decrease[8,16]. The gray matter volume of the prefrontal cortex reaches a plateau at 6 M in marmoset[30] and at puberty in humans[31]. Analysis of blood testosterone levels also revealed elevated levels in the perinatal period and puberty[32–34]. Taken together, 3 M in marmosets could correspond to 2–3 years in humans and 6 M to puberty, with a close developmental stage at birth for both species. To clarify the synaptic phenotype of the VPA-induced ASD marmoset model during synaptogenesis and pruning, we analyzed the excitatory synaptic structure and currents in layer 3 pyramidal neurons of Brodmann area 8b/9 from VPA-exposed (VPA) and unexposed (UE) marmosets (Fig. 1a)[35]. We focused on layer 3 pyramidal neurons given the relative enrichment of ASD-related gene expression in glutamatergic neurons in the superficial cortical layers in humans[27]. Accordingly, we made whole-cell recordings from layer 3 pyramidal neurons in acute slices obtained from UE and VPA marmosets at three timepoints. Furthermore, we morphologically analyzed the neurons after biocytin staining.

The density of dendritic spines on the basal dendrites changed with age. Specifically, it increased from 0 to 3 M and then decreased from 3 to 6 M in UE animals, which indicates a synaptic production overshoot followed by pruning (Fig. 1b, c). Compared to UE animals, the spine density in VPA animals was significantly lower at 0 M and higher at 6 M, but comparable at 3 M (Fig. 1c, number of samples in Supplementary Data 1). Although the data were not balanced with respect to sex, analysis using the data exclusively from males also showed a significant difference between the UE and VPA conditions at 0 M ($p = 0.0017$) and 6 M ($p = 0.00031$), suggesting that the imbalance in the sex ratio is not the cause of the difference. Among

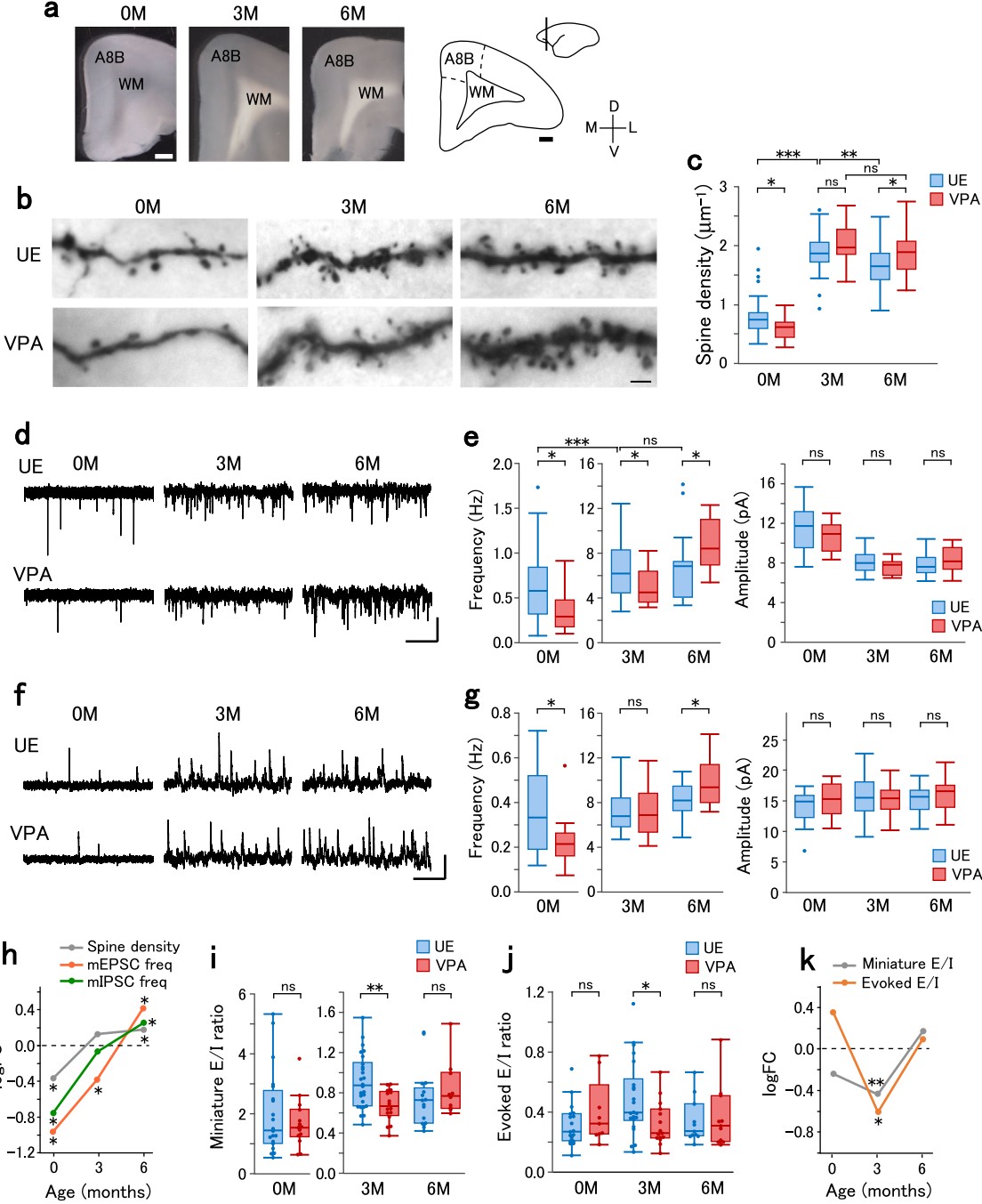

VPA animals, there was no significant difference in spine density at 3 M and 6 M, which suggests that synaptic pruning is compromised. The spine density was affected along the entire length of the apical and basal dendrites (Supplementary Fig. 2), indicating that the observed effect was common to both dendrites and not due to selective sampling. The elevated spine density at the puberty stage is consistent with human ASD[5,29] but not consistent with those in previous studies on rodent VPA models[36]. In contrast, there was no significant effect of VPA exposure on dendritic length (Supplementary Fig. 1).

Given that dendritic spines are sites of excitatory synapses, spine density abnormalities in VPA animals could indicate corresponding changes in excitatory synaptic connections. Therefore, we analyzed miniature excitatory postsynaptic currents (mEPSCs) in acute slices (Fig. 1d). The mEPSCs were blocked by

2,3-dioxo-6-nitro-1,2,3,4-tetrahydrobenzo[f]quinoxaline-7-sulfonamide (NBQX), which confirms that these currents are mediated by AMPA receptors (Supplementary Fig. 3). The mEPSC frequency changed with age (Fig. 1e, left). In UE animals, the mEPSC frequency increased from 0 M to 3 M; however, it did not significantly differ between 3M- and 6M-old animals. VPA exposure significantly altered the age-dependent change; specifically, the mEPSC frequency was significantly lower at 0 M and 3 M and higher at 6 M in VPA animals. There were no between-group differences in the mEPSC amplitude (Fig. 1e, right). These findings demonstrate a transition of VPA-induced alterations in the spine density and mEPSC frequency from underdevelopment at 0 M to overdevelopment at 6 M, as the log fold change (logFC, base-2 logarithm of the ratio of the value in VPA animals to the value in UE animals) plot shows (Fig. 1h).

**Fig. 1 Early underdevelopment and late overdevelopment of excitatory synapses in VPA-exposed marmosets. a** Slice preparations from UE animals at 0, 3, and 6 M, with schematic adapted from the atlas[64] on the right. Scale bar, 1 mm. A8B area 8b, WM white matter, M medial, L lateral, D dorsal, and V ventral. **b** Photomicrograph of biocytin-stained dendritic spines in layer 3 pyramidal neurons from UE and VPA animals. Scale bar: 2 μm. **c** Dendritic spine density in the basal dendrites of layer 3 pyramidal neurons from UE (blue) and VPA (red) animals. The spine density was measured at the dendritic segment 25–50 μm (0 M) or 50–75 μm (3 M and 6 M) from the soma. The boxplots represent the median, quartiles, and data range. $n = 37$ dendrites in 5 animals (0 M UE), $n = 22$ dendrites in 3 animals (0 M VPA), $n = 37$ dendrites in 6 animals (3 M UE), $n = 21$ dendrites in 3 animals (3 M VPA), $n = 38$ dendrites in 4 animals (6 M UE), and $n = 35$ dendrites in 3 animals (6 M VPA). Two-sided $t$-test with Holm-Sidak correction in UE animals between 0 M and 3 M, $p = 1.6 \times 10^{-21}$; between 3 M and 6 M, $p = 0.0053$; VPA animals between 3 M and 6 M, $p = 0.081$; between UE and VPA, $p = 0.015$ (0 M), $p = 0.095$ (3 M), $p = 0.019$ (6 M). ***$p < 0.001$; **$p < 0.01$; *$p < 0.05$; ns not significant. **d** Representative traces of mEPSCs recorded from layer 3 pyramidal neurons. Vertical scale bar, 10 pA. Horizontal scale bar, 2.5 s for 0 M and 0.5 s for 3 M and 6 M. **e** The frequency (left) and amplitude (right) of mEPSCs in UE and VPA animals. The boxplots represent the median, quartiles, and data range. $n = 20$ cells in 3 animals (0 M UE), $n = 14$ cells in 3 animals (0 M VPA), $n = 28$ cells in 6 animals (3 M UE), $n = 18$ cells in 4 animals (3 M VPA), $n = 21$ cells in 4 animals (6 M UE), and $n = 13$ cells in 3 animals (6 M VPA). Two-sided $t$-test with Holm-Sidak correction for the frequency in UE animals between 0 M and 3 M, $p = 4.5 \times 10^{-12}$; between 3 M and 6 M, $p = 0.99$; between UE and VPA, $p = 0.035$ (0 M), $p = 0.042$ (3 M), $p = 0.042$ (6 M). For the amplitude between UE and VPA, $p = 0.36$ (0 M), $p = 0.088$ (3 M), $p = 0.36$ (6 M). ***$p < 0.001$; *$p < 0.05$; ns not significant. **f** Representative mIPSC traces recorded from layer 3 pyramidal neurons. Vertical scale bar, 20 pA. Horizontal scale bar, 2.5 s for 0 M and 0.5 s for 3 M and 6 M. **g** The frequency (left) and amplitude (right) of mIPSCs in UE and VPA animals. The boxplots represent the median, quartiles, and data range. $n = 21$ cells in 3 animals (0 M UE), $n = 16$ cells in 3 animals (0 M VPA), $n = 37$ cells in 7 animals (3 M UE), $n = 25$ cells in 4 animals (3 M VPA), $n = 25$ cells in 5 animals (6 M UE), and $n = 16$ cells in 3 animals (6 M VPA). Two-sided $t$-test with Holm-Sidak correction for the frequency between UE and VPA, $p = 0.015$ (0 M), $p = 0.56$ (3 M), $p = 0.030$ (6 M). For the amplitude between UE and VPA, $p = 0.48$ (0 M), $p = 0.89$ (3 M), $p = 0.48$ (6 M). *$p < 0.05$; ns not significant. **h** logFC values of the spine density and the mEPSC and mIPSC frequencies, plotted as a function of age. *$p < 0.05$ (statistical tests and $p$ values are as in the legends of **c**, **e**, and **g**). **i** The E/I ratio of miniature synaptic currents for UE and VPA animals. The boxplots represent the median, quartiles, and data range. $n = 20$ cells in 3 animals (0 M UE), $n = 14$ cells in 3 animals (0 M VPA), $n = 26$ cells in 6 animals (3 M UE), $n = 16$ cells in 4 animals (3 M VPA), $n = 17$ cells in 4 animals (6 M UE), and $n = 10$ cells in 3 animals (6 M VPA). Two-sided $t$-test with Holm-Sidak correction between UE and VPA, $p = 0.62$ (0 M), $p = 0.0024$ (3 M), $p = 0.62$ (6 M). **$p < 0.01$; ns not significant. **j** The E/I ratio of the evoked synaptic currents. The boxplots represent the median, quartiles, and data range. $n = 19$ cells in 6 animals (0 M UE), $n = 9$ cells in 2 animals (0 M VPA), $n = 22$ cells in 5 animals (3 M UE), $n = 15$ cells in 2 animals (3 M VPA), $n = 11$ cells in 2 animals (6 M UE), and $n = 11$ cells in 2 animals (6 M VPA). Two-sided $t$-test with Holm-Sidak correction between UE and VPA, $p = 0.54$ (0 M), $p = 0.044$ (3 M), $p = 0.54$ (6 M). *$p < 0.05$; ns not significant. **k** logFC values of the E/I ratio in the frequency of miniature synaptic currents and the evoked synaptic currents, plotted as a function of age. **$p < 0.01$ and *$p < 0.05$ (statistical tests and p-values are as in the legends of **i** and **j**).

**Reduced E/I ratio in VPA-exposed marmoset infants.** The excitation-inhibition (E/I) balance is a regulator of the critical period[37] and an abnormal E/I balance is considered an ASD hallmark[38]. To analyze the E/I balance in the marmoset model, we recorded miniature inhibitory postsynaptic currents (mIPSCs) (Fig. 1f). The mIPSCs were blocked using picrotoxin, which confirms that these currents are mediated by $GABA_A$ receptors (Supplementary Fig. 3). In UE animals, the mIPSC frequency increased with age. Compared with UE animals, VPA animals had a significantly lower and higher mIPSC frequency at 0 and 6 M, respectively. However, it was comparable to that in UE animals at 3 M (Fig. 1g, left). There was no between-group difference in the mIPSC amplitude (Fig. 1g, right). The logFC plot showed a similar time course for the spine density and the frequencies of mEPSCs and mIPSCs (Fig. 1h). The slightly different time courses between the spine density, mEPSC frequency, and mIPSC frequency may reflect differences in the efficacy of synaptic transmission and the E/I ratio, as stated in the next section.

The ratio of the mESPC to mISPC frequency (the E/I ratio of miniature synaptic currents) was affected by VPA exposure (Fig. 1i). Specifically, this ratio was significantly lower in VPA animals at 3 M but not at 0 M or 6 M. To determine whether there were alterations in the E/I ratio of stimulus-evoked synaptic transmission, we recorded EPSCs and IPSCs from layer 3 pyramidal neurons in response to field stimulation of layers 4–5 (Supplementary Fig. 4a). As expected, VPA animals showed a significantly lower E/I ratio of evoked synaptic currents at 3 M but not at 0 M or 6 M (Fig. 1j). The age-dependent modulations of the E/I ratio are shown in the logFC plot (Fig. 1k). The time course of modulation was in contrast to the synaptic development (Fig. 1h) and suggests a distinct type of phenotype. The presynaptic properties of excitatory synapses were unlikely to be affected, because VPA exposure did not significantly affect the paired-pulse ratio of EPSCs (Supplementary Fig. 4b).

**Abnormal synaptic plasticity in VPA-exposed marmoset infants.** The E/I balance in cortical circuits is finely tuned by synaptic plasticity[39]. Since the frequencies of mEPSCs and mIPSCs in VPA animals at 3 M were lower than and comparable to those in UE animals, respectively, the altered E/I ratio in VPA animals observed at 3 M was presumably due to abnormal regulatory mechanisms of excitatory synapses. Previous studies on several rodent models of ASD have reported that long-term depression (LTD) of excitatory synapses is affected[40]. To determine whether VPA exposure affects LTD, we recorded the field EPSPs in layer 3 evoked by layer 4–5 stimulation and induced LTD by low-frequency stimulation (LFS, 1 Hz, 15 min). VPA exposure affected LFS-induced LTD in an age-dependent manner. At 0 M, LFS induced similar LTD magnitudes in UE and VPA animals. At 3 M, LFS induced LTD in VPA animals, but not in UE animals. At 6 M, LFS did not induce LTD in either UE or VPA animals (Fig. 2a, b).

LFS-induced LTD could be dependent on NMDA receptors; LTD at 0 M was blocked by D-2-amino-5-phosphonovaleric acid (D-APV) (Supplementary Fig. 5), although we did not test the NMDA dependence at 3 M due to limited availability of samples. Metabotropic glutamate receptor (mGluR)-dependent LTD was induced by (S)-3,5-dihydroxyphenylglycine (DHPG), which is a group I mGluR agonist that induces LTD in the rat visual cortex[41], to a comparable magnitude in UE and VPA animals at 0 M. mGluR-dependent LTD was not induced in either UE or VPA animals at 3 M (Supplementary Fig. 5). Therefore, prenatal VPA exposure does not affect mGluR-dependent LTD.

LTD is accompanied by dendritic spine shrinkage in rodent hippocampal neurons[42]. Since different levels of LTD may lead to distinct distributions of spine sizes, we analyzed the dendritic spine volume (Supplementary Fig. 6). Compared with UE animals, VPA animals showed significantly lower spine volume at 3 M, but the spine volume in VPA animals was comparable to UE animals at 0 M and 6 M (Fig. 2c). Consistently, electron

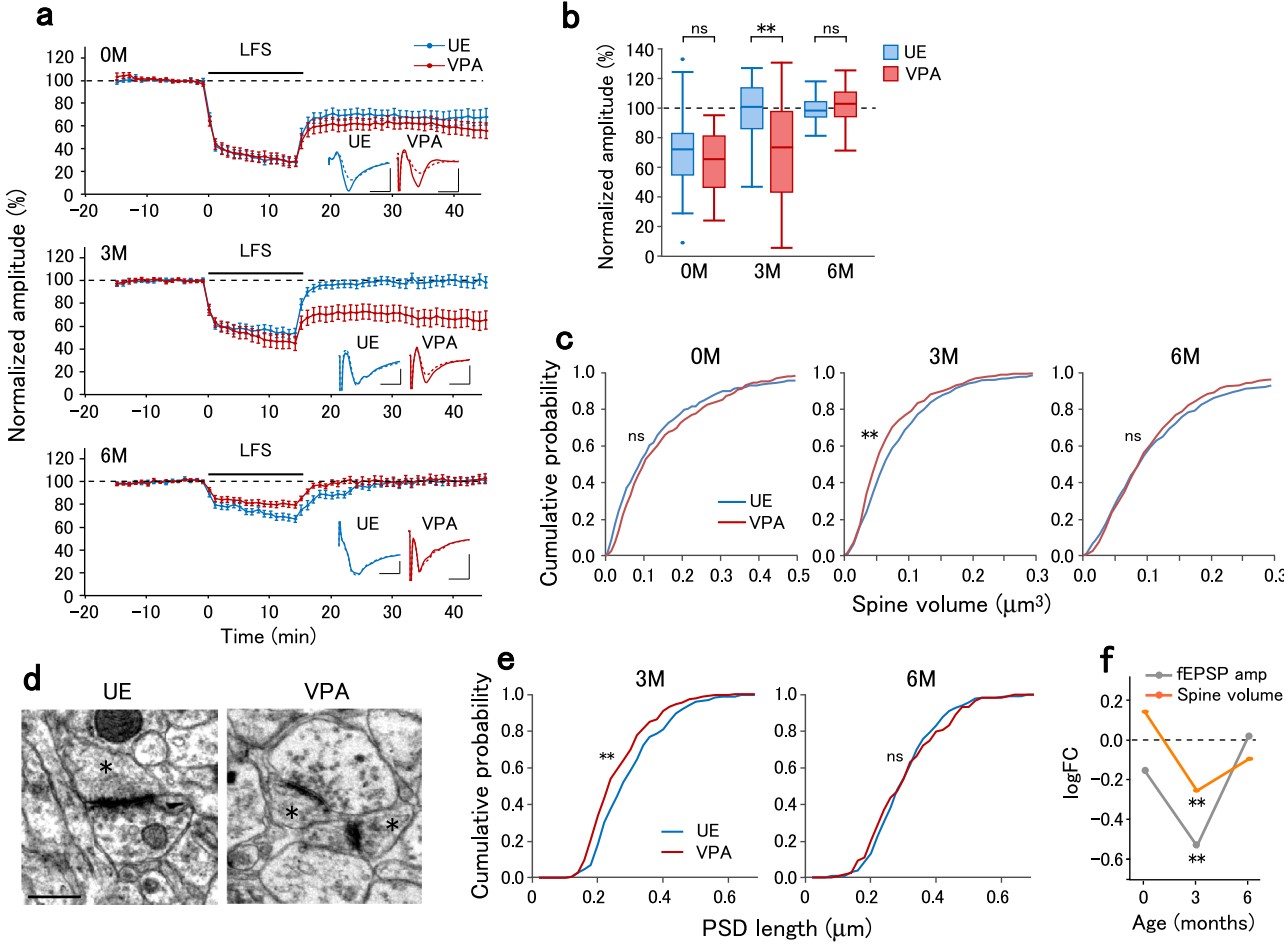

**Fig. 2 Age-dependent modification of LTD and spine volume in VPA-exposed marmosets. a** The time course of normalized field EPSP amplitudes before and after LFS onset that were recorded from slices obtained at 0 M (top), 3 M (middle), and 6 M (bottom) in UE (blue) and VPA (red) animals. Error bars represent the SEM. Insets represent the average traces of field EPSPs before (−10 to 0 min, solid lines) and after (30 to 40 min, dotted lines) LFS. $n = 26$ pathways in 12 animals (0 M UE), $n = 15$ pathways in 2 animals (0 M VPA), $n = 20$ pathways in 5 animals (3 M UE), $n = 22$ pathways in 5 animals (3 M VPA), $n = 16$ pathways in 2 animals (6 M UE), and $n = 15$ pathways in 1 animal (6 M VPA). **b** Normalized field EPSP amplitudes in UE and VPA animals at post-LFS 30–40 min. The boxplots represent the median, quartiles, and data range. Two-sided $t$-test with Holm-Sidak correction between UE and VPA, $p = 0.38$ (0 M), $p = 0.0020$ (3 M), $p = 0.76$ (6 M). **$p < 0.01$; ns, not significant. **c** Cumulative distribution of spine volume in UE and VPA animals at 0, 3, and 6 M. $n = 227$ spines in 3 animals (0 M UE), $n = 291$ spines in 2 animals (0 M VPA), $n = 467$ spines in 5 animals (3 M UE), $n = 399$ spines in 2 animals (3 M VPA), $n = 472$ spines in 6 animals (6 M UE), and $n = 406$ spines in 3 animals (6 M VPA). Two-sided Kolmogorov–Smirnov test between UE and VPA animals, $p = 0.089$ (0 M), $p = 0.0025$ (3 M), $p = 0.64$ (6 M). **$p < 0.01$; ns not significant. **d** Representative electron micrograph of synapses in UE and VPA animals at 3 M. These images are representative of 234 PSDs (UE) and 230 PSDs (VPA). Asterisks indicate dendritic spines. Scale bar, 0.5 μm. **e** Cumulative distribution of the PSD length in UE and VPA animals at 3 M and 6 M. $n = 234$ PSDs in 2 animals (3 M UE), $n = 230$ PSDs in 2 animals (3 M VPA), $n = 236$ PSDs in 2 animals (6 M UE), and $n = 119$ PSDs in 1 animal (6 M VPA). Two-sided Kolmogorov–Smirnov test between UE and VPA animals at 3 M, $p = 0.0026$; 6 M, $p = 0.69$. **$p < 0.01$; ns not significant. **f** logFC values for the normalized post-LFS field EPSP amplitudes and the spine volume plotted as a function of age. **$p < 0.01$ (statistical tests and $p$ values are as in the legends of **b** and **c**).

microscopy confirmed VPA exposure-induced synaptic structure alterations at 3 M (Fig. 2d). Compared with UE animals, VPA animals had a significantly lower length of postsynaptic density (PSD) in layer 3 of the dorsomedial prefrontal cortex at 3 M, but not at 6 M (Fig. 2e). The logFC values for both the field EPSP amplitude after LFS and the average spine volume showed a reduction specifically at 3 M (Fig. 2f).

In summary, prenatal VPA exposure affects the E/I ratio, LTD, and spine volume at 3 M with the effects at other ages being weaker. This suggests that 3 M is another timepoint with strong synaptic phenotypes.

**Altered vocalization in VPA-exposed marmoset infants**. We postulated that early synaptic development could affect social

behavior at an early age. Although adult VPA marmosets show abnormal social behavior[17,18], the behavior of infantile animals has not been studied. In humans, language deficits with ASD become evident in childhood. The dorsomedial prefrontal cortex is among the brain regions connected to the presupplementary motor area involved in vocalization in primates[43,44]; therefore, early synaptic abnormalities in the prefrontal cortex may affect vocal development.

We recorded the calls of marmosets around 3 M in an isolated environment for 5 min and annotated each call (Fig. 3a). The total call number during isolation tended to be lower in VPA animals; however, there was no significant difference (Fig. 3b). UE marmosets produced various calls, with phee calls being the most frequent and constituting 44% of all calls. VPA animals produced a higher ratio of phee calls (79%), with lower

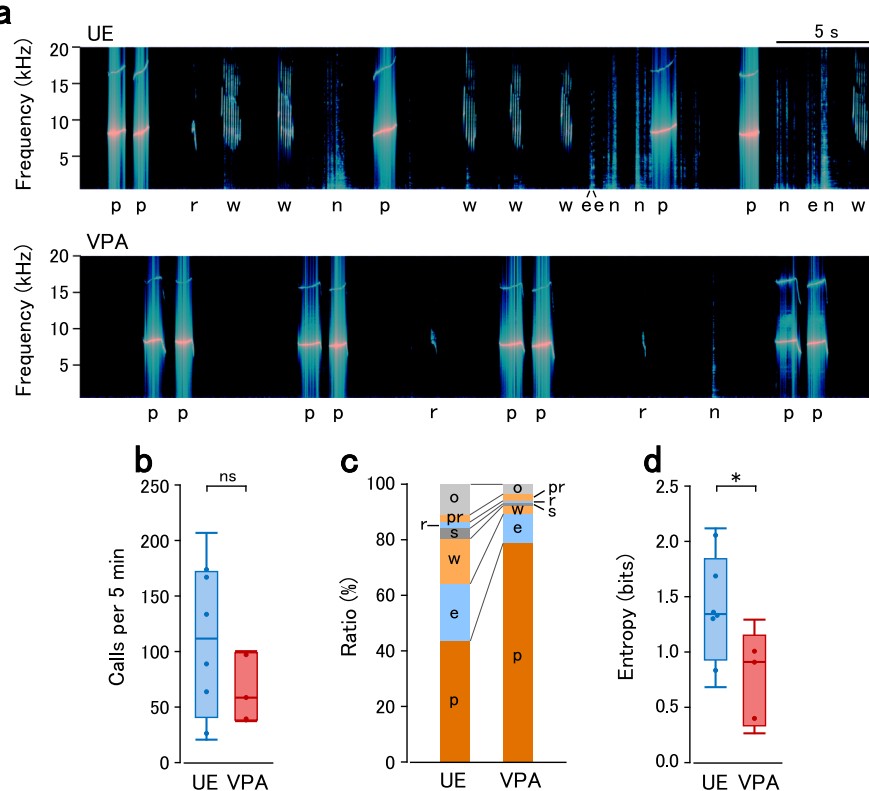

**Fig. 3 Altered isolation calls in VPA-exposed marmosets. a** Representative vocalization spectrogram for UE (top) and VPA (top) animals at 12 weeks of age. The type of call is shown below the spectrogram. Call types: e ekk or cough, p phee, r trill, w twitter, n indicates noise. **b** The average number of calls in UE and VPA animals at 11–13 weeks. The boxplots represent the median, quartiles, and data range. $n = 8$ animals (UE) and $n = 5$ animals (VPA). Two-sided $t$-test, $p = 0.16$. ns, not significant. **c** Average ratio of call types in UE and VPA animals. Call types: pr phee-trill or trill-phee, s tsik, o others. Other annotations are as in (**a**). **d** The average entropy of calls in UE and VPA animals. The boxplots represent the median, quartiles, and data range. Two-sided $t$-test, $p = 0.036$. *$p < 0.05$.

frequencies for other call types (Fig. 3c). The predominance of the phee call in VPA animals led to a significantly lower entropy of the call types in VPA animals than in UE animals (Fig. 3d). The behavioral abnormality found in infancy may be the consequence of abnormal synaptic development between birth and infancy.

**Altered gene expression in VPA-exposed marmosets.** We observed two types of time courses of synaptic phenotypes, one being most affected at 0 M (Fig. 1h) and the other most affected at 3 M (Figs. 1k and 2f). We speculated that these phenotypes are accompanied by altered gene expression. We searched for VPA-modulated gene expression using a custom-made microarray. Among 9362 genes expressed in the three cortical regions (areas 8, 12, and TE), there were 1037 differentially expressed genes (DEGs) with an absolute value of logFC of >0.4 and an adjusted $p$-value ($p_{\mathrm{adj}}$) of <0.05 at either age.

Since there were synaptic phenotype alterations with age, we analyzed the temporal profile of gene expression alterations. The temporal profile of logFC values revealed distinct VPA-induced modulations. Using the $k$-means clustering method, the DEGs were grouped into three clusters as indicated by the colored dots (Fig. 4a). These clusters were discernibly separated on the plot of logFC values at 3 M against those at 0 M. There was no significant overlap of upregulated or downregulated genes between 0 and 3 M or 0 and 6 M. In contrast, the overlap between 3 and 6 M was significantly high (Fig. 4b), although more DEGs were present at 3 M than at 6 M.

All logFC values of cluster 1 genes were substantially low at 0 M (Fig. 4c). The logFC values of cluster 2 and 3 genes were negative

and positive at 3 M, respectively, while those at 0 M and 6 M were smaller in magnitude (Fig. 4c). The genes with the strongest expression modulations are shown in Fig. 4d. We also provided a comprehensive list of genes in each cluster (Supplementary Data 2). The time course of the logFC values for cluster 1 genes corresponded to that of the spine density and mEPSC and mIPSC frequencies, as indicated by the low mean distance between the logFC values of the synaptic parameters and gene expression modulations (Fig. 4e). This suggests that cluster 1 includes genes associated with abnormal synaptic development. The time course of the logFC values for cluster 2 genes was consistent with those of the E/I ratios, LTD (post-LFS field EPSP amplitudes), and spine volume. Although the time course of the logFC values for cluster 3 genes was not correlated with synaptic modulations, the negative of the logFC values of cluster 3 genes was parallel to the logFC of the E/I ratios, post-LFS field EPSP amplitudes, and spine volume ("3r" in Fig. 4e). These results suggest that clusters 2 and 3 include genes associated with synaptic modulations that specifically occur at 3 M.

Cluster 1, but not clusters 2 and 3, was significantly enriched with ASD-associated (SFARI) genes (Fig. 4f), which include a number of synapse-related genes. Furthermore, we analyzed critical period-related gene expression, since critical period plasticity in sensory areas is affected in ASD model animals[45]. Cluster 2, but not clusters 1 and 3, was enriched with critical period-related genes that had the highest expression at the critical period in the visual cortex of rodents[46,47] (Fig. 4f, Supplementary Data 3). Pathway analysis revealed that the synaptogenesis signaling pathway was among the highly affected pathways of clusters 1 and 3, and cholesterol biosynthesis-related pathways were among the highly affected

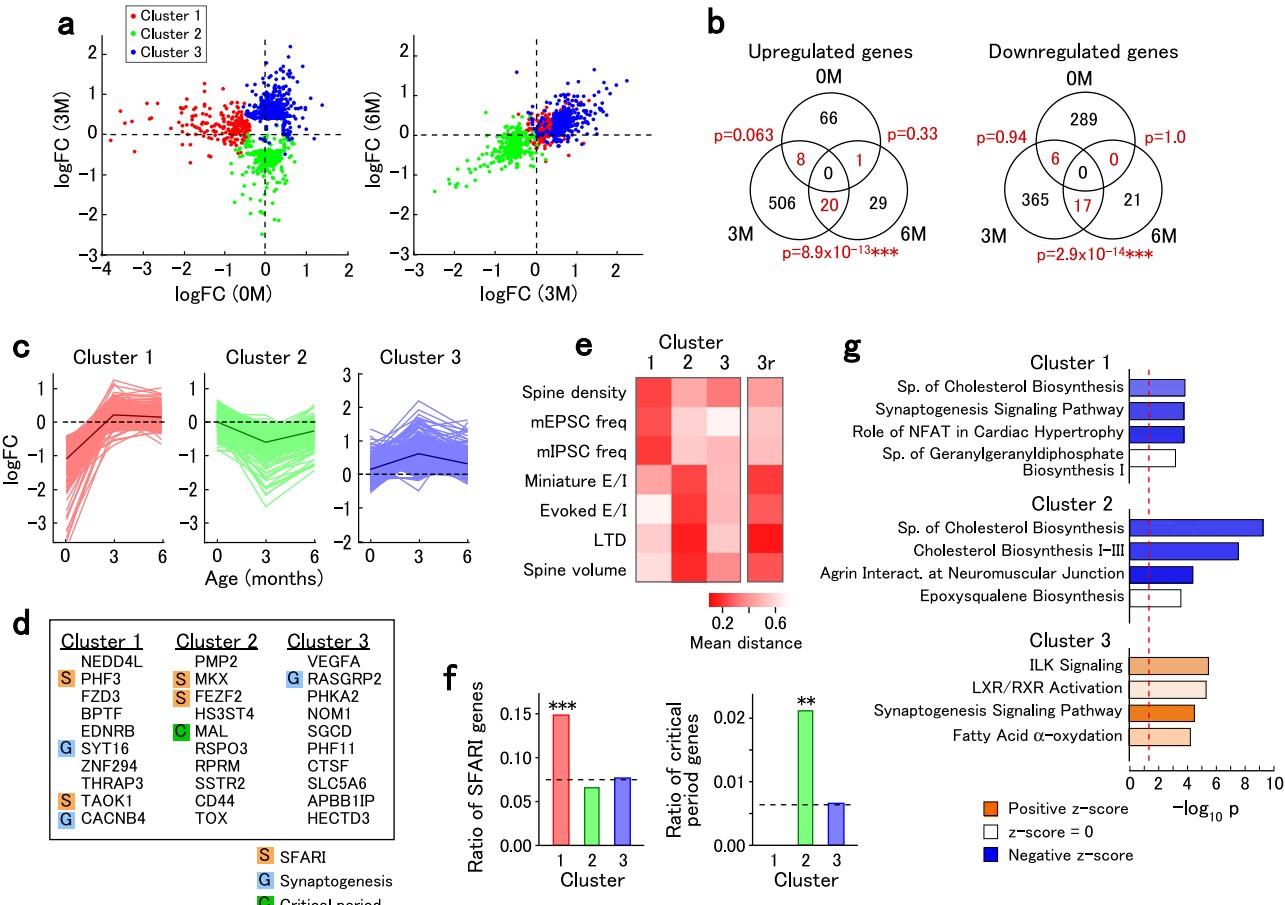

**Fig. 4 Microarray analysis of differential gene expression. a** Distribution of the logFC values of gene expression significantly modulated by VPA exposure. The logFC values at 3 M are plotted against those at 0 M (left) while the logFC values at 6 M are plotted against those at 3 M (right). Each gene is colored based on the cluster. The microarray data are from 10 samples from 5 animals (0 M UE), 11 samples from 4 animals (0 M VPA), 9 samples from 6 animals (3 M UE), 6 samples from 2 animals (3 M VPA), 9 samples from 6 animals (6 M UE), and 6 samples from 2 animals (6 M VPA). **b** Common upregulated and downregulated genes across ages. *P* values are for the enrichment of common genes between ages (one-sided Fisher's exact test). **c** The time course of logFC values. The traces in light colors represent data from individual genes in the cluster. The traces in dark colors represent the average of all genes in the cluster. **d** List of genes with the highest expression modulations in each cluster. SFARI, synaptogenesis-related, and critical period-related genes are marked. **e** The distance between the average logFC values of gene expression for each cluster and the logFC values of structural and physiological parameters (spine density, mEPSC frequency, mIPSC frequency, E/I ratio of miniature synaptic currents, E/I ratio of evoked synaptic currents, post-LTD field EPSP amplitudes, and spine volume). The column labeled "3r" shows the distance between the negative of the average logFC values of cluster 3 and the structural and physiological parameters to highlight parameters that are inversely correlated with cluster 3. **f** Enrichment of SFARI and critical period-related genes in each cluster. The dotted line is the ratio among the entire gene set. One-sided Fisher's exact test for SFARI gene enrichment, $p = 0.00088$ (0 M), $p = 0.90$ (3 M), $p = 0.68$ (6 M); for critical period gene enrichment, $p = 1.0$ (0 M), $p = 0.0028$ (3 M), $p = 0.57$ (6 M). ***$p < 0.001$, **$p < 0.01$. **g** Enriched pathways for each cluster. The color of the bars represents the direction of regulation of the pathway based on the logFC values at 0 M (cluster 1) and 3 M (clusters 2 and 3). Sp, superpathway. The p-values of enrichment were provided by the IPA software. The red dotted line represents the threshold of significance ($p = 0.05$).

pathways of clusters 1 and 2 (Fig. 4g; Supplementary Data 4). As suggested by the small overlap of gene expression modulations between 0 M and 3 M, upregulated and downregulated genes in the synaptogenesis signaling pathway were distinct between these timepoints (Supplementary Fig. 7). These findings suggest that genes affected by VPA exposure at different times were probably involved in different aspects of synaptic modulations in ASD. Using upstream analysis, we searched potential drugs to normalize the altered biological functions at 0 M and 3 M (Table 1). At 0 M, statins and an anti-inflammatory drug were predicted. At 3 M, various drugs that affect myelin, inflammation and calcium signaling were predicted.

Finally, we compared gene expression modulations in VPA-exposed marmosets with those in human ASD, which were from postmortem brains of mostly juvenile to adult individuals with idiopathic ASD[48] (Fig. 5a). There was a significant positive correlation of gene expression modulations at 3 M and 6 M in marmosets with those in human ASD, although fewer genes were modulated at 6 M than at 3 M. In contrast, there was no correlation between the gene expression modulations at 0 M and those of human ASD (Fig. 5a), which could be because human ASD samples were from juvenile to adult individuals. Gene expression modulations in the marmoset model at 3 M and human ASD were further compared for each gene module constructed by weighted gene coexpression network analysis (WGCNA) on typically developed and ASD human samples[48]. In both the marmoset model and human ASD, genes in modules associated with neurons and oligodendrocytes were downregulated, while genes in modules associated with astrocytes and microglia were upregulated (Supplementary Fig. 8). Gene

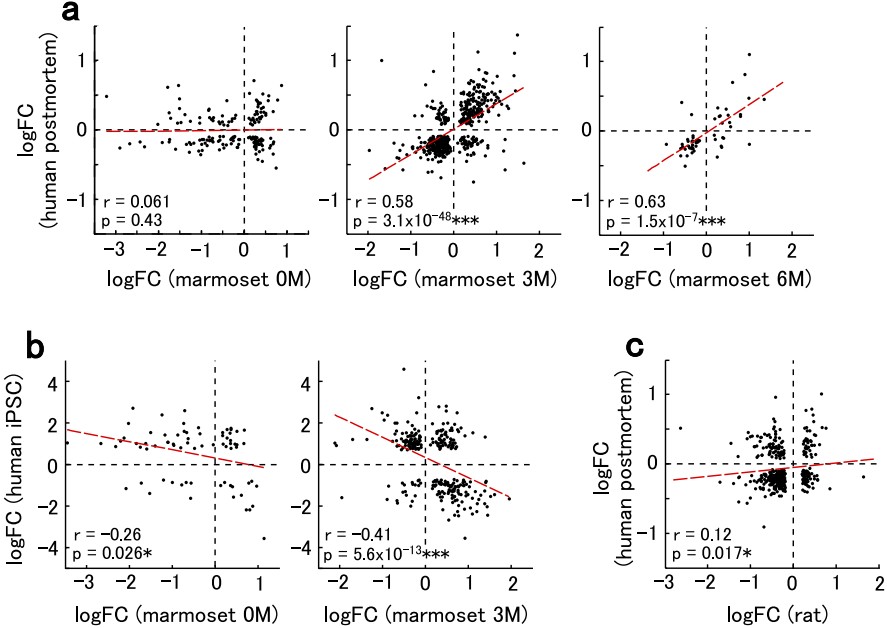

**Fig. 5 Relationship between gene expression modulations in VPA-exposed marmosets and human ASD. a** Relationship between the logFC values in marmosets at 0 M (left), 3 M (center), and 6 M (right), and postmortem human ASD[48]. Modulated genes with $p_{adj} < 0.1$ are plotted. For each plot, Spearman's correlation coefficients ($r$) and two-sided $p$-values ($p$) are shown. The dashed red line is the linear regression. ***$p < 0.001$. **b** Relationship between the logFC values in marmosets at 0 M (left) and 3 M (right) and iPSC-derived neurons from human ASD (31 days of terminal differentiation)[50]. Modulated genes with $p_{adj} < 0.1$ are plotted. Spearman's correlation coefficients ($r$) and two-sided $p$-values ($p$) are shown. Dashed red line is the linear regression. ***$p < 0.001$, *$p < 0.05$. **c** Relationship between the logFC values in VPA-exposed rats[51] and postmortem human ASD. Modulated genes with $p_{adj} < 0.1$ are plotted. Spearman's correlation coefficients ($r$) and two-sided $p$-values ($p$) are shown. Dashed red line is the linear regression. *$p < 0.05$.

expression modulations in neurons derived from induced pluripotent stem cells (iPSCs) from people with ASD, which represents the mid-fetal stage, should be inversely modulated from those observed in postmortem samples[49]. Indeed, we observed a negative correlation between the gene expression modulations in marmosets at 3 M and those in iPSC-derived neurons from humans with ASD[50] (Fig. 5b). Again, the correlation between marmosets at 0 M and iPSC-derived neurons was lower (Fig. 5b). These results suggest that gene expression modulations at the perinatal stage are different from those at both the fetal and mature stages.

The correlation between VPA-exposed marmosets (3 or 6 M) and postmortem human ASD samples was higher than the correlation between the rat VPA model at postnatal day 35[51] and the human samples (Fig. 5c). The VPA-exposed rats showed concordant changes with human ASD in modules associated with neurons and oligodendrocytes, but not in modules associated with astrocytes and microglia (Supplementary Fig. 8). In some modules, gene expression was modulated in opposite directions between the rat model and human ASD. Furthermore, various mouse models of ASD exhibited fewer concordant modules associated with limited cell types (Supplementary Fig. 8). These data suggest that VPA-exposed marmosets replicate a broad range of human ASD components, while rodent models generally replicate a part of the pathology.

## Discussion

The present study revealed the existence of distinct synaptic phenotypes with two types of time courses in the ASD marmoset model. Spine density and miniature synaptic current frequency were underdeveloped at the neonatal stage and overdeveloped in puberty. In contrast, the E/I ratio and LTD were affected in infancy. The cortical transcriptome revealed three gene clusters with related time courses of modulation with the synaptic

phenotypes (Fig. 6). These results may suggest the importance of age-specific treatment for ASD. Moreover, there was a good correlation of post-infancy gene expression alterations with human ASD, suggesting a common affected developmental trajectory between the marmoset model and human ASD.

In this study, we examined layer 3 pyramidal neurons in area 8b/9 at 3 M (infancy) along with 0 M (neonate) and 6 M (puberty). This is because the spine density in layer 3 pyramidal neurons in marmoset area 8b/9 reaches a peak at 3 months, which coincides with ~2–3 years in humans when the spine density in the medial prefrontal layer 3 neurons also reaches a peak[7,28]. Symptoms of ASD become apparent and diagnosable before 3 years, and treatment starting around or before this time is desirable. Our study also showed the presence of ASD-like behavior in the marmoset model at 3 months, in addition to the previously found phenotypes in adults[17,18], indicating the importance of infantile ages in ASD pathogenesis. Although area 8b/9 is thought to play a major role in the pathogenesis of ASD, it is important to pay attention to the developmental patterns of other areas because they may also contribute to ASD symptoms. In particular, marmoset studies have suggested that cortical maturation proceeds in a caudo-rostral fashion[30,52,53], as in humans[7]. Although the spine density in both area 8b/9 and the primary visual cortex of marmosets peaks at 3 months[8,9,16], the spine density declines more slowly in the prefrontal cortex than in the primary visual cortex. The delay in synaptic pruning in the prefrontal cortex suggests that circuit maturation continues longer in area 8b/9. This is also consistent with the magnetic resonance imaging study that showed a delayed onset of decline in the cortical thickness in the prefrontal cortex compared to the primary visual cortex (11 months vs 6 months)[30]. The molecular maturation of pyramidal neurons also varies between layers, and for layer 3 prefrontal neurons, the final stage of refinement does not occur until 12 months[52]. This suggests that the phenotypes

**Table 1 Drugs from upstream analysis that are predicted to normalize gene expression modulation at 0 and 3 M.**

| Age | Drug | z-score | p value of overlap | Target molecules |
|---|---|---|---|---|
| 0 M | Rosuvastatin | −2.236 | 1.5E-04 | DHCR7, FDFT1, FDPS, HMGCR, IDI1 |
| 0 M | Methylprednisolone | −2.224 | 6.7E-03 | ADCY6, ASNS, EDNRB, FBXO32, FDFT1, IGF1, IL6ST, INSIG1, ITGB1, MAPK14, NOLC1, PEX11A, SLC16A1, SLC7A2, TNKS2, TYRO3, XRCC1 |
| 0 M | Lovastatin | −2.176 | 6.4E-04 | ACAT2, ADAM10, DHCR7, FAM189B, FDFT1, FDPS, HMGCR, IGF1, POLG |
| 0 M | Atorvastatin | −2.086 | 2.3E-03 | DHCR24, DHCR7, FDFT1, FDPS, HMGCR, HSD17B7, IDI1, SMAD2, SQLE, VEGFA |
| 3 M | Triptolide | −3.300 | 2.4E-05 | ADM, AKAP12, BMP6, CADM1, DPP4, DUSP1, ENG, FBLN5, HPGD, IGFBP2 |
| 3 M | Ursolic acid | −2.764 | 1.0E-02 | CCND1, MAL, MBP, MYC, PLP1, PTGS2, SLC38A2, VCAM1 |
| 3 M | Halofuginone | −2.433 | 1.4E-04 | ADD3, AKAP12, CCN2, CD44, CTSK, CTSS, FBLN5, FSTL1, GNG11, GPX3 |
| 3 M | Aspirin | −2.431 | 1.2E-05 | ABCA1, ADM, AKAP12, BMP6, CADM1, CCND1, CTSD, DPP4, ENG, FBLN5 |
| 3 M | Caffeic acid phenethyl ester | −2.394 | 1.1E-02 | ABCB1, CCND1, CD44, F3, FCGRT, GNAI2, HSPB1, PTGS2, VCAM1, VIM |
| 3 M | Losartan potassium | −2.318 | 1.8E-05 | CCN2, CREM, ENG, F3, FN1, ITGB5, ITPR2, PDGFA, POSTN, PTGS2 |
| 3 M | Metformin | −2.213 | 4.5E-02 | ABCA1, ABCB1, CCN2, F3, FN1, ILK, MYC, PTGS2, PYGM, RARRES2 |
| 3 M | Verapamil | −2.213 | 7.6E-03 | ABCB1, ANXA2, KCNC1, S100A10, TXNIP, VCAM1 |
| 3 M | Obeticholic acid | −2.177 | 4.2E-02 | ABCB1, CCN2, FN1, PTGS2, TIMP1 |
| 3 M | Actinomycin D | −2.173 | 4.6E-03 | ABCA1, ADM, ARC, CASP9, CCN2, CCND1, DUSP1, F3, FEN1, FN1 |
| 3 M | Curcumin | −2.093 | 7.6E-03 | ABCA1, ABCB1, APOE, AXL, CCN2, CCND1, CD44, DIABLO, F3, GSTP1 |
| 3 M | Puerarin | −2.000 | 7.2E-04 | ABCB1, CASP9, PTGS2, TXN |
| 3 M | Paroxetine | −2.000 | 1.5E-02 | ABCA1, ADM, ANKRD37, PDK1 |

The z-score is for the predicted activation state of the drug. The p value of overlap is the probability that the targets of the drug and the modulated genes coincide. No drug was predicted from the 6 M data.

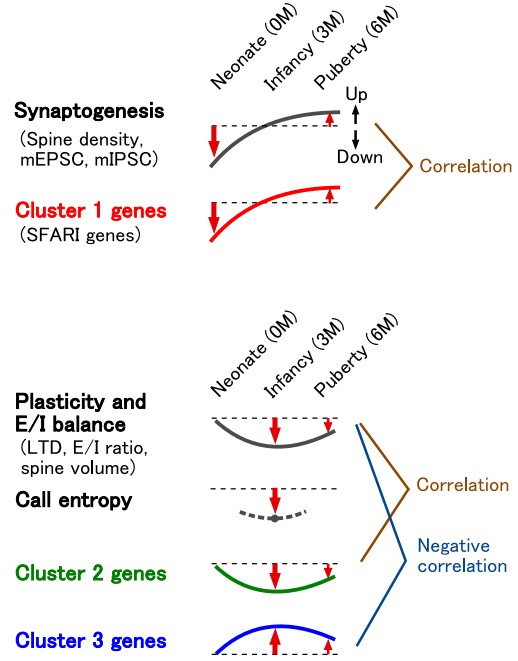

**Fig. 6 Summary of the results.** Synaptogenesis-related phenotypes, including modulations in spine density and miniature synaptic current frequencies, were correlated with modulations in the expression of cluster 1 genes, which include SFARI genes. The E/I balance and plasticity-related phenotypes, including modulations in the E/I ratio, LTD (post-LFS field EPSP), and spine volume, were correlated with modulations in cluster 2 genes and inversely correlated with modulations in cluster 3 genes. The entropy of calls was reduced at infancy when cluster 2 and 3 genes were maximally modulated.

after complete maturation of the circuit may be different from the phenotypes we observed until puberty; moreover, different therapeutic approaches may be needed for infantile and adult people with ASD. Future research on phenotypes at later times into adulthood and in different areas or layers will provide further insight into ASD pathology.

Previous studies have reported abnormal dendritic spine development in human ASD[4,5,29]. Although the spine density was significantly higher in people with ASD at 13–19 years of age, there was no significant difference observed at the age of 2–9 years[5]. Consistently, we observed an elevated spine density in VPA-exposed marmosets at 6 M. In contrast, the spine density in VPA-exposed marmosets was lower at 0 M, indicating a distinct synaptic phenotype at the neonatal stage. Cluster 1 genes were downregulated in neonates; additionally, the temporal modulation pattern of cluster 1 genes was similar to that of the spine density and the frequency of miniature synaptic currents. This suggests an association of altered gene expression with abnormal synaptic development. Indeed, the synaptogenesis signaling pathway was among the highly affected pathways for this cluster. Cluster 1 was enriched with SFARI genes including *SETD5* (SET domain containing 5), which is involved in DNA methylation and implies an epigenetic mechanism in the pathogenesis of ASD[54].

We observed a reduced E/I ratio in VPA animals at 3 M. The E/I ratio is a common circuit property that is affected in human ASD and animal models[38]. LTD was also enhanced in VPA animals at infancy. Correspondingly, genes in clusters 2 and 3 were modulated maximally at 3 M. The E/I balance regulates homeostatic synapse tuning during circuit remodeling[39], and is an essential factor for the critical period plasticity of ocular dominance in the visual cortex[37]. Cluster 2 included genes that show the highest expression in the critical period (~4 weeks of age) in rodents[46,47], and some genes in clusters 2 and 3 have been implicated in synaptic regulation around this timepoint, including *CADM1* (cell adhesion molecule 1, cluster 3), *LRRTM4* (leucine-rich repeat transmembrane 4, cluster 2)[55], and *GRIN2A* (glutamate ionotropic receptor NMDA type subunit 2A, cluster 2)[56]. These synaptic and molecular phenotypes are consistent with the critical period theory of ASD[45]. Abnormal expression of the NMDA receptor subunits *GRIN1* (glutamate ionotropic receptor NMDA type subunit 1, cluster 3) and *GRIN2A* may explain the age-dependent modulation of LTD.

Most people with ASD have difficulty in context-dependent communication and often produce stereotyped repetitive speech during infancy. An increased phee call ratio and decreased vocalization variability were also observed in VPA-exposed

marmosets at infancy. Early experience is necessary for the development of language in humans[26] and vocalization in marmosets[57]. Vocalization disturbance may be the consequence of early synaptic and molecular phenotypes, suggesting under-development of circuits at the neonatal stage and abnormal network plasticity at infancy in the marmoset model. In particular, the critical period-related gene *CADM1* is involved in the development of vocalization[58].

Gene expression alterations at 3 M and 6 M showed a positive correlation with those in human postmortem samples of idiopathic ASD from juvenile to adult ages (Fig. 5). The marmoset model replicated human gene expression modulations in nine modules associated with four cell types (neurons, oligodendrocytes, astrocytes, and microglia) (Supplementary Fig. 8). In contrast, various rodent ASD models, including drug-induced, inbred, and genetically modified models, replicated human modulations in only a part of these modules and fewer cell types. In the rat VPA model, only six modules were concordant with human ASD, and those associated with astrocytes and microglia were not among the concordant modules. Genetically modified mouse models had even fewer concordant modules with human ASD. For example, the *Fmr1* knockout mouse had an astrocyte-related module that tended to be modulated in the same direction as human ASD, and the *Mecp2* heterozygous mouse had a neuron/oligodendrocyte-related module modulated in the same direction with human ASD, but none of these modules were significantly concordant with human ASD. The lower concordance between rodents and humans is consistent with a greater divergence of gene coexpression networks, especially in glial-cell-related modules, between rodents and humans than between non-human primates and humans[13]. The replication of gene expression modulations in glia-related modules in the marmoset model seems particularly important, since ASD is thought to be caused by a mechanism involving glial regulation of synaptic formation and function[59].

Considering the theory that early impairment in the development of ASD (e.g., reduced attention to the face) can lead to later impairment in higher social skills (sophisticated face recognition)[60], early intervention based on early biological evidence is highly desirable. Pharmacological interventions after adolescence generally have short-term effects. Normalizing the molecular functions at appropriate timepoints until infancy, together with behavioral intervention, could be an effective ASD therapy. The upstream analysis revealed candidate drugs for normalizing the molecular phenotypes. At 0 M, statins were predicted to affect cholesterol biosynthesis, which was down-regulated in VPA animals (Fig. 4g). Cholesterol, which is a component of lipid rafts and a regulator of synaptic functions, has been implicated in human ASD and ASD model animals[61]. Lovastatin, a statin drug that affects cholesterol synthesis, has effects on rat model of fragile-X syndrome[62]. At 3 M, ursolic acid was among the predicted drugs. Ursolic acid has anti-inflammatory effects and promotes myelination[63], which is implicated in the closure of the visual critical period[25] and may help stabilize synapses.

The present study revealed that VPA model marmosets exhibit synaptic, behavioral, and molecular phenotypes close to those of human ASD. In particular, the larger number of concordant gene modules between this model and human ASD suggests that it is more appropriate as a model of human ASD than rodent models. Together with our previous studies on autistic behavior and morphological alterations of the brain[17,18,23,24], the marmoset ASD model appeared to be a useful model for translational study. However, the present study has the following limitations. First, individuals with idiopathic ASD have different etiologies and exhibit different symptoms. While the synaptic and molecular

phenotypes in the ASD marmoset model after infancy revealed in this study were similar to the overall conditions of people with idiopathic ASD, the translational validity of this model will be further enhanced by determining the subtype that this model best represents. Second, we did not address the roles of non-neuronal components such as astrocytes and microglia, of which accumulating evidence suggests involvement in the pathogenesis of ASD. Third, early-age intervention proposed in this study requires early screening and diagnosis which are not fully established, and medication to neonates and children generally involves the risk of adverse side effects. Nonetheless, this study indicated the critical importance of early biological phenotypes in ASD. These findings could contribute to future age-dependent therapy for ASD in early life, which could be highly efficient.

## Methods

**Animals.** All experiments were approved by the Animal Research Committee of the National Center of Neurology and Psychiatry and the Animal Care and Use Committee of the National Institute of Radiological Sciences and were in accordance with the NIH Guide for the Care and Use of Laboratory Animals. Marmosets were bred in the National Center of Neurology and Psychiatry or the National Institute of Radiological Sciences, kept under a 12 h/12 h light/dark cycle, and provided with food (CMS-1, CLEA Japan) and water ad libitum. The temperature and humidity were maintained at 27–30 °C and 40–50%, respectively. Serum progesterone levels in female marmosets were measured; furthermore, we determined the ovulation time. VPA was prepared as a 4% solution in 10% glucose solution and intragastrically administered to pregnant marmosets daily for 7 days from post-conception day 60 at 200 mg/kg/day. Some UE animals received 10% glucose-solution, while other UE animals received no treatment. Untreated animals were used because of the limited availability of treated animals. The offspring of VPA-administered or UE marmosets were used at 0 M (4–11 days for electrophysiological and morphological analyses, 0–4 days for molecular analyses), 3 M (88–110 days for electrophysiological and morphological analyses, 84–99 days for molecular analyses), and 6 M (189–273 days for electrophysiological and morphological analyses, 180–208 days for molecular analyses). There were no between-group differences in the pregnancy periods of the glucose-administered UE and VPA marmosets (139.5 ± 1.2 days for glucose-administered UE [mean ± SD, $n = 6$] and 140.6 ± 1.7 days for VPA [$n = 16$], $p = 0.10$ with two-sided $t$-test), as well as the body weight at birth (29.0 ± 1.9 g for glucose-administered UE and 30.7 ± 2.4 g for VPA, $p = 0.088$). There was no significant difference in the body weight at birth between glucose-administered and untreated UE animals (29.7 ± 2.9 g for untreated animals, $n = 36$, $p = 0.45$).

**Slice preparation and electrophysiology.** The marmosets were deeply anesthetized using ketamine hydrochloride (20 mg/kg, i.m.) and sodium pentobarbital (100 mg/kg, i.p.). Next, they were transcardially perfused with ice-cold $CO_2/O_2$-saturated artificial cerebrospinal fluid (ACSF); subsequently, the skull was removed and the brain was isolated. The ACSF comprised the following (in mM): 126 NaCl, 3 KCl, 1.2 $NaH_2PO_4$, 10 glucose, 26 $NaHCO_3$, 2.4 $CaCl_2$, and 1.3 $MgSO_4$. Coronal slices containing the dorsomedial prefrontal cortex (area 8b/9) were prepared using a vibratome (model MT, Dosaka EM) at 400 μm thickness (Fig. 1a). Typically, 7–8 slices were obtained from each hemisphere. The slices were placed on an interface-style chamber perfused with ACSF at 32 °C to allow recovery.

The location of area 8b/9 at 3 M and 6 M was identified based on the atlas[64], and further confirmed in by the cytoarchitecture (thick layer 3, moderately developed layer 4, and homogeneous layer 5)[35]. Cells in layer 5 in area 9 were smaller than those observed in area 8b. The cytoarchitecture at 0 M has not been documented, but it showed a similar cytoarchitecture to 3 M and 6 M, with more densely packed cell bodies. We obtained whole-cell recordings from layer 3 pyramidal neurons under an infrared differential image contrast (IR-DIC) microscope (BX51WI, Olympus). The layer with relatively sparse pyramidal neurons up to 1 mm from the pia was identified as layer 3. The layer structure was confirmed by Nissl staining (Supplementary Fig. 1). The slices were placed on a recording chamber that was continuously perfused with ACSF at 28 °C. To record miniature synaptic currents, 1 μM tetrodotoxin (Fujifilm Wako Pure Chemical) was added to the ACSF. In some experiments, the AMPA receptor blocker NBQX (Abcam, 20 μM) or the GABA$_A$ receptor blocker picrotoxin (Cayman Chemical, 100 μM) was added to the ACSF. The internal solution contained the following (in mM): 130 Cs methanesulfonate, 10 NaCl, 5 $MgSO_4$, 10 HEPES, 0.6 EGTA, 2 Na-ATP, 0.6 Na-GTP, 10 Na-phosphocreatine, and 3 mg/ml biocytin. A patch-clamp amplifier (Axopatch 1D, Axon Instruments) was used for signal recording with a low-pass filter of 2 kHz. The signals were recorded on a PC using a data acquisition board (PCI-6221, National Instruments) and Igor Pro software (WaveMetrics, version 6.0) at a 10 kHz sampling frequency. Data with unstable baseline or with a series resistance >28 MΩ (0 M) or >18 MΩ (3 and 6 M) were excluded. We

recorded mEPSCs at −65 mV, which was close to the calculated equilibrium potential for Cl⁻, and mIPSCs at 0 mV, which was close to the equilibrium potential for monovalent cations. Miniature synaptic currents were analyzed using Mini Analysis (Synaptosoft, version 6.03) and a custom-made program for MATLAB (Mathworks, versions 7.9 and 9.10). The signal was further bandpass filtered at 4–1000 Hz. Moreover, the standard deviation (SD) of the baseline was measured in a 50 ms segment lacking events. We set the current amplitude threshold (in units of SD of the baseline) and current area threshold (in units of ms times SD of the baseline) for miniature events at 3 and 4.5 (0 M mEPSCs), 3 and 9 (0 M mIPSCs), 2 and 3 (3 M and 6 M, mEPSCs), 2 and 6 (3 M and 6 M, mIPSCs), respectively.

Stimulus-evoked synaptic currents were induced by field stimulation of layers 4–5 using a bipolar tungsten electrode (tip separation 150 μm) connected to an isolator (BSI-950, Dagan). The stimulus duration was 0.1 ms. The intensity was set to 300 μA, which evoked approximately half-maximal EPSCs. EPSCs and IPSCs were recorded at holding potentials of −65 and 0 mV, respectively.

Stimulus-evoked field potentials were recorded in an interface-style recording chamber under a binocular microscope. The chamber was maintained at 35 °C and a glass recording electrode containing 0.5 M NaCl was placed in layer 3 at ~400–500 μm from the pia. A bipolar tungsten stimulation electrode was placed in layers 4–5 at ~1–1.2 mm from the pia. The signal was amplified using an amplifier (ER-1, Cygnus) with a low-pass filter at 1 kHz and recorded on a PC at a 10 kHz sampling frequency. Test stimuli were applied at a 30 s interstimulus interval. Up to two pathways were used in the same slice. The stimulus duration was 0.1 ms. The stimulus intensity was adjusted to evoke a field EPSP with one phase decay (100–500 μA). The field EPSP was completely abolished after perfusion with 20 μM NBQX (Tocris) and 50 μM D-APV (Cayman Chemical). When the baseline amplitude was stable for >20 min, we applied LFS (1 s interval, 900 times) from the stimulation electrode. The dependency of LFS-induced LTD on NMDA receptors was confirmed using 100 μM D-APV in 0 M slices. We induced mGluR-dependent LTD by perfusion with 100 μM DHPG (Tocris) for 10 min.

**Neuron structure analysis.** After whole-cell recording using biocytin-containing electrodes, the slices were fixed in 4% paraformaldehyde overnight and stained using the Vectastain Elite ABC Kit (Vector Laboratories) and DAB Substrate Kit (Vector Laboratories). The slices were dehydrated using a graded ethanol series, cleared with xylene, and mounted with Entellan New (Merck).

The neuron structure was analyzed using Neurolucida (MBF Bioscience, versions 10 and 11). The whole dendritic arbor was traced using a ×20 objective. The dendritic spine density was analyzed using a ×100 oil immersion objective (NA 1.49) on the basal dendrites at 25–50 μm from the soma for 0 M and at 50–75 μm from the soma for 3 M and 6 M, which corresponded to the segments of highest spine density[16]. We also measured spine density along the entire length of basal and apical dendrites. We used an intensity-based method to calculate the spine volume given that direct measurement of the spine diameter is unreliable for small spines. Axial and transverse profiles of 3–6 medium-sized spine heads on a dendrite were fitted using a Gaussian curve (Supplementary Fig. 6). For spines with a diameter >0.4 μm, twice the standard deviation of the curve (2σ) matches the actual spine head diameter observed using an ultrahigh voltage electron micrograph (Oga, T. and Fujita, I., unpublished observation). Therefore, we used 2σ as the diameter. The spine volume was calculated using the transverse ($d_t$) and axial ($d_a$) diameters as $\pi d_t^2 d_a/6$. We obtained a proportional relationship between the calculated spine volume and optical density for spine heads used as references (diameter > 0.4 μm and volume < 0.4 μm³) (Supplementary Fig. 6). Using this relationship, the other spine volumes were estimated by their optical density measurements.

For Nissl staining, the fixed slices were cryosectioned at 40 μm, and after defatting with chloroform/ethanol and rehydration, the sections were stained with 0.1% thionin, dehydrated, cleared with xylene, and mounted.

For electron microscopy, slices were immediately fixed after slicing in 2.5% glutaraldehyde and 2% paraformaldehyde in 0.1 M cacodylate buffer (pH 7.2) for at least 5 days. The slices were then treated using 1% osmium oxide in cacodylate buffer for 60 min and stained with 2% uranyl acetate for 60 min. After serial dehydration, the tissues were embedded in Epon 812 (TAAB). Ultrathin sections (70 nm thick) were prepared and examined using a Tecnai Spirit electron microscope (Thermo Fisher Scientific-FEI).

**Analysis of vocalization.** Marmoset vocalizations were recorded using a linear PCM recorder (Olympus, LS-100). The marmosets were isolated in a soundproof chamber and were allowed habituation to the environment for ~1 min before each recording session. The animals were placed in a small recording cage (38 × 43 × 47 cm³) located 10 cm apart from the recorder. Furthermore, vocalizations uttered during 5-min recording sessions were obtained in 24 bits at a 96-kHz sampling frequency. The recordings were started from postnatal weeks 1–3 and were performed every week up to 20 weeks, of which the data from weeks 11–13 were analyzed. The data were analyzed by an experimenter who did not know the type of animal using the Praat software (https://www.fon.hum.uva.nl/praat, version 6.1)[65]. Each call was annotated as one of the following: ekk or cough, phee, trill,

tsik, twitter, phee-trill or trill-phee, and other[66–68]. Continuous sequences of multiple calls without a silent gap (multisyllabic calls) were considered a single bout of call. Call type entropy was calculated using the ratio of each call type $r_i$ as $-\Sigma_i \, r_i \log_2 r_i$.

**Microarray analysis.** Microarray analysis was performed using brain samples from UE and VPA marmosets at 0, 3, and 6 M[69,70]. In addition to the two prefrontal areas (areas 8 and 12), we used the temporal association area (area TE) to increase the reliability of data. Marmosets were anesthetized using ketamine hydrochloride and sodium pentobarbital. Subsequently, they were transcardially perfused with diethylpyrocarbonate-treated phosphate-buffered saline followed by isolation of the cortical tissues and immersion in RNAlater (Thermo Fisher). Total RNA was extracted using the RNeasy Mini Kit (Qiagen). RNA integrity was assessed using a Bioanalyzer (Agilent Technologies) and samples with an RIN value > 7 were used. A biotin-labeled cRNA probe was generated using the GeneChip 3'IVT Express Kit (Affymetrix). The probe was hybridized with a custom-made microarray (Marmo2a520631F)[69,70] using the GeneChip Hybridization, Wash, and Stain Kit (Affymetrix). The microarray was scanned using the GeneChip Scanner 3000 (Affymetrix) and processed using MAS5 (https://www.bioconductor.org/packages/release/bioc/html/affy.html) to examine the reliability of probe detection. The data were normalized using GCRMA (https://www.bioconductor.org/packages/release/bioc/html/gcrma.html). We considered genes with a log₂ expression value > 5 as being expressed in the brain tissue. Protein-coding genes were selected based on the HUGO Gene Nomenclature Committee database (https://www.genenames.org). Differential gene expression was evaluated based on the $p$ value from Welch's $t$-test with Benjamini-Hochberg adjustment ($p_{adj}$). DEGs (absolute value of logFC > 0.4 and $p_{adj} < 0.05$ at either age) were clustered using the $k$-means algorithm. The cluster number was set to 3, which yielded the lowest value of Akaike's information criterion. For affected genes with multiple probes, we used data from the probe with the lowest $p_{adj}$.

The mean distance between the logFC of synaptic phenotypes (logFC$_{syn}$) and gene expression (logFC$_{gene}$) was calculated as $\Sigma_t \, |\text{logFC}_{syn}(t) - \text{logFC}_{gene}(t)|/3$, where $t$ is the age (0 M, 3 M, or 6 M). The list of ASD-related genes was derived from the Simons Foundation Autism Research Initiative (SFARI, http://www.sfari.org, release 08-07-2020). The list of critical period-related genes was obtained by combining data from two studies[46,47]. Pathway and upstream analyses were conducted using the IPA software (Qiagen, Summer Release 2020). The logFC values at 0 M were used for the pathway analysis of cluster 1, and those at 3 M were used for clusters 2 and 3. To predict drugs that potentially normalize gene expression modulations, upstream analysis was performed for the modulated genes at each age, and drugs and chemicals with predicted inhibition (activation z-score < −2 with no bias) were selected. To compare gene expression modulations in the marmoset model with those in human ASD (postmortem samples[48] or iPSC-derived neurons[50]), commonly modulated genes with $p_{adj} < 0.1$ were used. To compare gene expression modulations in a rat ASD model[51] with those in human ASD, gene symbols were converted using HomoloGene (NCBI, https://www.ncbi.nlm.nih.gov/homologene, release 68).

**Statistical analysis.** To compare UE and VPA data at multiple timepoints, the data were tested by Welch's $t$-test with Holm-Sidak correction. Two-way repeated-measures ANOVA was used to compare the spine density along the dendrites and the paired-pulse ratio. Comparisons between two groups were performed using the $t$-test. The Kolmogorov–Smirnov test was used to compare the distribution of spine volume and PSD length. Enrichment of genes was analyzed using Fisher's exact test. Spearman's correlation was used to determine the correlation of logFC values between marmosets or rats and humans. The concordance of gene expression modulations between the model animals and human ASD was tested using the binomial test. All the data were from distinct samples. Tests for gene enrichment and concordance were one-sided, and all other tests were two-sided.

**Reporting summary.** Further information on research design is available in the Nature Research Reporting Summary linked to this article.

## Data availability
The microarray data generated in this study have been deposited at NCBI GEO under accession number GSE156186. All other data supporting the findings of this study are in the Source Data file and Supplementary Data 1-4. All public domain data used in this study are listed in the Methods section. Source data are provided with this paper.

## Code availability
The source codes for the analysis of miniature synaptic currents and spine volume have been deposited at GitHub (https://github.com/ncnp-bisai/matlab).

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

## Acknowledgements

We thank A. Tsuchiya for assisting with the marmoset experiments. This work was supported by Intramural Research Grant for Neurological and Psychiatric Disorders from the National Center of Neurology and Psychiatry (29-6, N.I.), Novartis Research Grant 2019 (S.W.), JSPS KAKENHI Grant Number JP18K06497 (J.N.), and AMED Grant Number JP21dm0207066 (N.I.).

## Author contributions

S.W., T.K., T.O., J.N., and N.I. planned the project and designed the experiments. S.W., T.K., and T.O. performed electrophysiological analyses. J.N., R.I., and A.N. performed behavioral analyses. S.W., T.O., K. Sumida., K.H., K. Saito, and I.M. performed microarray analyses. K.N. and T.M. produced ASD model marmosets. K. Sakai performed electron microscopy analyses. M.S. and K.W. supported electrophysiological experiments. S.W. and N.I. wrote the manuscript. All authors discussed the results and commented on the manuscript.

## Competing interests

The authors declare no competing interests.
