## [Peer Review File · Nature Communications]

Reviewers' Comments:

Reviewer #1:

Remarks to the Author:

Review of Watanabe et al. Nat Commun

This paper reports on some structural and molecular characteristics of the marmoset cortex in a model of autism spectrum disorder. The paper is highly relevant because it helps argue that this model can be used for biomedical investigation of ways to mitigate the effects of this disease. The paper contains new data, which will be important for investigators in several fields of neuroscience. However, there are some issues that would need to be resolved before I can give it fully enthusiastic support.

Main issues:

1. Analysis of spine density – Spine density varies markedly with distance from the soma, and this has been confirmed for other cortical areas of the marmoset (Elston et al. 1999, JCN). However, one of the paper's main claims is based on measurements made at a specific distance from the cell body. What is more, the distance chosen for this measurement is different for the newborn versus 3 and 6 month old animals. This is simply not convincing. An analysis similar to the one presented in the Elston et al. paper should be included, in order to demonstrate that the trends observed in spine number are not just result of selective sampling.
2. The paper only looks at basal dendrites, but it seems (Fig. S1) that the neurons were sampled in coronal sections. Thus, there is no reason why a similar analysis could not be done for apical dendrites. This is important because some of the evidence that is used to justify the similarity between marmoset and humans also incorporates data from apical dendrites.
3. The paper ignores important contributions by Missler et al. (e.g. J Comp Neurol. 1993 Jul 1;333(1):53-67), which did careful work on the synaptogenesis of marmoset V1. Instead, most of the references to facts such as the overproduction and pruning of spines is based on mouse papers. This is unfortunate, particularly since comparisons between the prefrontal and visual areas raise some questions that need to be addressed in the discussion of this paper. For example, according to one of the most widely accepted views, cortical development would obey a caudal to rostral gradient (e.g. Sawiak et al. Cereb Cortex 2018 Dec 1;28(12):4440-4453. Thus, it is strange that the reported peak is the same (around 3 months) in V1 and in the medial prefrontal cortex.
4. Similarly, it would be useful to consider (at least, in the discussion) the evidence from molecular maturation that suggests that some areas of the prefrontal cortex are not really fully mature until after 12 months, particularly in layer 3 (Burman et al. Eur J Neurosci Mar;25(6):1767-79. How can this have impacted on the interpretation of the results?
5. The documentation of the area from where the samples were taken is not satisfactory. At the minimum, this needs to be improved by providing a supplementary figure showing how the slices used to get the samples at different ages compares with the Paxinos et al. 2012 atlas. Considering that different areas mature at different rates, it is important to reassure the reader that the samples that were actually used were obtained in corresponding locations, and that the areas that are claimed to be sampled correspond to those in the atlas.

Minor or specific issues/ suggestions:

6. The paper tries to adopt a translation-focused tone, but this is perhaps a bit overdone. For example, in the abstract (second sentence): "Early intervention immediately after diagnosis in childhood is desirable to improve prognosis". This is true, but creates an immediate expectation from the reader that this paper will deal with diagnosis of ASD, or remediation. The techniques used here cannot be used for diagnosis in humans. Perhaps consider here and in other parts the balance between pointing out the value of the information being presented, versus overdoing the medical angle.

7. Intro, "Normal synaptic development proceeds in a precisely regulated manner, where excessive synaptic formation in the early postnatal stage is followed by pruning (Huttenlocher and Dabholkar, 1997)" – consider citing papers that provide marmoset-based data here, as well.

8. Intro, "Moreover, the developmental expression pattern of some genes are primate-specific..." – IS primate specific

9. Intro, "The common marmoset (*Callithrix jacchus*), a small New World monkey, is well suited to studies on neurodevelopmental diseases, and previous studies from our laboratory have described similarities in cortical architecture and synaptic development to humans (Oga et al., 2013; Sasaki et al., 2015)". Both references, from the lab that also originates the present paper, deal with dendritic trees, not cortical architecture. This links to another problem that is brushed over in this paper, that is, the fact that the region from where the data were obtained is never documented, and there is little discussion of whether areas 8b and 9 in marmoset correspond to human areas. One possibility is to cite here papers that have made the case that the areas being sampled are similar in architecture and connections in marmosets and humans (e.g. Burman et al. 2006 *J Comp Neurol* 2006 Mar 10;495(2):149-72; Reser et al. 2013 *Cereb Cortex* 23(8):1901-22) and then document the slices relative to the Paxinos atlas, as suggested above. Otherwise, the similarity should be documented perhaps in supplementary figures.

10. Intro, "ages of birth (0M), 3 months (3M), and 6 months (6M), which can be taken to correspond to neonatal, infancy, and adolescence periods, respectively" – this is questionable. IN the Sawiak et al. paper (see above) it is proposed that marmoset adolescence is between 8 and 20 months (please note that sexual maturity is usually achieved by 12-13 months). Thus, the present paper may in fact be dealing with pre-pubertal marmosets even at the oldest time point. The developmental time line proposed by Sawiak also seems to be more in line with the Burman et al. 2007 results (both papers deal with cortical development, and address areas 8 and 9).

11. Results, "Therefore, 3M in marmosets could correspond to ~3 years in humans" – this is close, but according to Sawiak 3M would correspond to a younger human. Not sure this makes a difference, but maybe allow for the possibility (e.g. 2-3 years).

12. Results, "cortical pyramidal neurons of Brodmann area 8b/9" – state layer (L3 presumably) and how the areas were identified.

13. Results, vocalization: "the dorsomedial prefrontal cortex is among the brain regions involved in vocalization in primates (Miller et al., 2010)". This statement, as presented, is slightly misleading. The Miller paper does not identify areas 8 and 9 as being the places where c-fos labeled cells were found. Instead they state: "Based on descriptions from previous research (Brodmann, 1909; Burman et al., 2006), the two most rostral regions in our analysis (Frontal Regions 1 and 2) showing high levels of cFos expression likely correspond to ventral-prefrontal cortex area 12/45, whereas the populations in the most caudal region (Frontal Region 3) are likely premotor cortex areas 6V and 6D. Future research will aim to refine this analysis and ascertain the functional role and anatomical boundaries of these neuronal populations".

14. Results, vocalization: "These findings are indicative of the existence of behavioral abnormality in VPA animals following abnormal synaptic development between birth and infancy". The conclusion here amounts to a generic statement, to the effect that something changes in the vocalization, and something changes in the brain, so the two must be related. I don't think that the paper presents sufficient evidence to establish a causal link between the two sets of observations. This can be presented as another evidence of ASD-like symptoms, but try not to stretch too far.

15. The discussion should mention whether marmosets and humans are born at corresponding developmental stages, or if not which species is more immature at birth. The paper reports differences between marmosets and humans at birth, and knowing the developmental stages would be important to interpret the significance of this finding. If I understand correctly the discussion on gene changes does allow for the possibility that 0M marmoset corresponds to a

younger human stage, but not for the possibility that marmosets are born more mature than humans.

Reviewer #2:

Remarks to the Author:

This study by Watanabe and colleagues treats marmosets with valproic acid as a model of ASD. The authors then carry out a number of electrophysiological, neuroanatomical, behavioral, and genomic experiments. In general, the individual experiments are mostly carried out appropriately and results are described well. I found it intriguing that the marmoset genomic datasets overlapped better with the human ASD datasets than the rat valproic acid model, supporting the use of the marmoset. However, I think the manuscript suffers a bit in organization of how the data are integrated across the three time points and discussed. Furthermore, these new results could be better discussed within the context of the previous literature. At the end of the paper, it is challenging to have a clear take away message with specific targets or time points that underlie the changes in synaptic physiology. This is mostly because results vary widely across the three time points depending on the experiments. Some results are significant and others are not, depending on age and/or experiment. Perhaps the authors need to make a schematic chart that integrates all the data at each time point, or maybe even one or two of the time points should be eliminated.

In addition to this "tightening" of the story, I have some other minor suggestions:

1. The use of the specific time points should probably reference McKinnell et al., 2001.
2. The authors mentioned in the animals' section that some animals in the UE group received 10% glucose (vehicle), which was used to make 4% VPA solution. Why did they give this vehicle to only some animals and not to all?
3. In few of their experiments, the number of samples for 0M and 6M are many fewer than the 3M group. Any specific reason for this?
4. I could not understand or find a good explanation for the synaptic changes in different parameters in one group and not in other.
5. The authors find changes in the synaptic structure with VPA treatment only in the 3M group. Did they check in other groups as well?
6. The authors used DAPV in that 0M group to show that it blocks NMDA dependent LFS-induced LTD and say that probably NMDA dependent LTD is enhanced in the 3M group. However, they should do the same DAPV blockage in the 3M group and then show that after blockage with DAPV, there is no difference in the LTD between VPA and UE animals to confirm that the change in LTD is NMDA dependent.
7. In the introduction, the authors claim 85% of ASD is idiopathic with no genetic cause, but do not provide a citation to back that claim. I believe there are many genetic studies now showing that the percentage of known genetic causes is significantly higher than 15%.
8. In Fig 1g, at the 6 month timepoint, the logfold change of the 3 time points relative to one another is different but the authors do not comment on this.
9. In Fig 2d, the authors do EM of synapses at 3M. Have the other 2 time points been examined?
10. In the discussion, the authors use the term "mature" several times. But, I assume, they really mean adolescent?
11. In the discussion, the authors mention LTD is required for vocal learning in songbirds. Since the 3M VPA monkeys have enhanced LTD, how do the authors reconcile this with the abnormal vocalizations?

Reviewer #3:

Remarks to the Author:

Watanabe et al
NCOMMS-20-40776

This paper reports the results of studies in the valproic acid (VPA) model of autism in the marmoset. Three lines of research are described. 1) Electrophysiological studies of in vitro slices

taken from the frontal cortex 2) vocalization studies of the living animals and 3) gene expression analyses of tissue taken from both frontal and temporal lobes. Most of this work was done in marmosets who were newborn, 3-months-old or 6-months-old.

Critique:

This is an ambitious series of studies based on the premise that the nonhuman primate is more appropriate for modeling autism than rodent models. A second premise is that the VPA model is a valid model for human autism.

The paper is not well written so it is difficult to follow the flow of arguments presented by the authors. It is curious that the rationale for conducting the three levels of investigation outlined above was not presented in this paper. Neither does the paper adequately make the case that the NHP provides more useful data than the many dozens of rodents VPA studies.

There are also many statements made throughout the paper that are not substantiated by the literature. For example, "Synaptic overdevelopment in the mature marmoset model was in parallel to human ASD." This statement has no references to back it up and as far as I know, this "fact" has not yet been established using human tissue."

I congratulate the authors on attempting a very complex series of studies. But, I would encourage them to recast this paper in a more focused manner. E/I balance, language development and gene expression do not necessarily fit in the same paper. I would also encourage a more accurate representation of the autism literature. Much of what is presented as well-established fact is based on initial or unreplicated studies. A more circumspect approach to representing the state of autism research would be very welcome.

Point-by-point response to the reviewers' comments

Reviewers' comments are shown as blue text. Changes to the manuscript are highlighted in yellow.

Reviewer #1 (Remarks to the Author):

This paper reports on some structural and molecular characteristics of the marmoset cortex in a model of autism spectrum disorder. The paper is highly relevant because it helps argue that this model can be used for biomedical investigation of ways to mitigate the effects of this disease. The paper contains new data, which will be important for investigators in several fields of neuroscience. However, there are some issues that would need to be resolved before I can give it fully enthusiastic support.

We are pleased to hear that the reviewer found the study important for the field. We appreciate the comments from an expert with in-depth knowledge of the marmoset brain. We have carefully revised the manuscript according to the comments. We are very grateful for the comments, as we feel they helped us to significantly improve the manuscript.

Main issues:

1. Analysis of spine density – Spine density varies markedly with distance from the soma, and this has been confirmed for other cortical areas of the marmoset (Elston et. al. 1999, JCN). However, one of the paper's main claims is based on measurements made at a specific distance from the cell body. What is more, the distance chosen for this measurement is different for the newborn versus 3 and 6 month old animals. This is simply not convincing. An analysis similar to the one presented in the Elston et al. paper should be included, in order to demonstrate that the trends observed in spine number are not just result of selective sampling.

We thank the reviewer for the valuable suggestion. We have measured the spine density along the entire length of basal dendrites as done by Elston et al. (1999), and clarified that age-dependent modulation in spine density was not due to selective sampling. The results have been presented in Suppl. Fig. 2 and in the text (page 6, line 13). We feel that the addition of the data according to the reviewer's suggestion has made the results much more solid and reliable.

2. The paper only looks at basal dendrites, but it seems (Fig. S1) that the neurons were sampled

in coronal sections. Thus, there is no reason why a similar analysis could not be done for apical dendrites. This is important because some of the evidence that is used to justify the similarity between marmoset and humans also incorporates data from apical dendrites.

We again thank the reviewer for the suggestion to analyze apical dendrites. We have measured the spine density along apical dendrites, and clarified similar modulation of spine density as in basal dendrites. The results have been presented in Suppl. Fig. 2 and in the text (page 6, line 13).

3. The paper ignores important contributions by Missler et al. (e.g. *J Comp Neurol.* 1993 Jul 1;333(1):53-67), which did careful work on the synaptogenesis of marmoset V1. Instead, most of the references to facts such as the overproduction and pruning of spines is based on mouse papers. This is unfortunate, particularly since comparisons between the prefrontal and visual areas raise some questions that need to be addressed in the discussion of this paper. For example, according to one of the most widely accepted views, cortical development would obey a caudal to rostral gradient (e.g. Sawiak et al. *Cereb Cortex* 2018 Dec 1;28(12):4440-4453. Thus, it is strange that the reported peak is the same (around 3 months) in V1 and in the medial prefrontal cortex.

We thank the reviewer for the information of previous marmoset studies, which we have now cited in the revised manuscript (page 5, line 14). We have carefully compared the results of previous studies on neurodevelopment in V1 and the prefrontal cortex of marmoset and humans. Some studies reported regional differences in synaptic development, while others did not. The electron microscopy study by Huttenlocher and Dabholkar (1997) described a relatively large difference between the peak times of synaptic density between V1 and the prefrontal cortex in humans. In contrast, Bakken et al. (2016) reported similar peak times for both areas in humans and macaque. Sasaki et al. (2015) also showed similar peak times of spine density in marmoset. In addition, Sawiak et al. (2018) reported only slightly different development of gray matter volume between these areas. These studies have been summarized and described in the Results section (page 5, line 14). The regional difference in synaptic development is an important issue that requires further consideration, and since we analyzed synaptic development in just one area, we discussed it as a limitation of this study in the Discussion section (page 18, line 2).

4. Similarly, it would be useful to consider (at least, in the discussion) the evidence from molecular maturation that suggests that some areas of the prefrontal cortex are not really fully mature until after 12 months, particularly in layer 3 (Burman et al. *Eur J Neurosci*

Mar;25(6):1767-79. How can this have impacted on the interpretation of the results?

We thank the reviewer for the information about the Burman et al. (2007) paper, which provides us with the timeline of neuronal maturation across cortical layers. As the reviewer pointed out, Burman et al. (2007) suggested that marmoset is not fully mature at the oldest age point we studied; moreover, we analyzed only layer 3 neurons, which exhibit relatively slow maturation. The absence of adult data and the limited area/layer are limitations of this study, and we discussed these limitations in the Discussion section (page 18, line 2).

5. The documentation of the area from where the samples were taken is not satisfactory. At the minimum, this needs to be improved by providing a supplementary figure showing how the slices used to get the samples at different ages compares with the Paxinos et al. 2012 atlas. Considering that different areas mature at different rates, it is important to reassure the reader that the samples that were actually used were obtained in corresponding locations, and that the areas that are claimed to be sampled correspond to those in the atlas.

We thank the reviewer for the important comment on the cortical area we studied. We have selected the area based on the Paxinos atlas, and we have made a figure that compares the slice we used with the atlas (Suppl. Fig. 1). We hope that this will help the readers understand from which area we obtained the data.

Minor or specific issues/ suggestions:

6. The paper tries to adopt a translation-focused tone, but this is perhaps a bit overdone. For example, in the abstract (second sentence): “Early intervention immediately after diagnosis in childhood is desirable to improve prognosis”. This is true, but creates an immediate expectation from the reader that this paper will deal with diagnosis of ASD, or remediation. The techniques used here cannot be used for diagnosis in humans. Perhaps consider here and in other parts the balance between pointing out the value of the information being presented, versus overdoing the medical angle.

We agree with the reviewer’s comment that the translational value of the study was overstated in the manuscript, although our final goal is translation to human ASD. We have removed the word “diagnosis” from the Abstract, and avoided the excessive use of the word “translational” in the Discussion.

7. Intro, “Normal synaptic development proceeds in a precisely regulated manner, where excessive synaptic formation in the early postnatal stage is followed by pruning (Huttenlocher and Dabholkar, 1997)” – consider citing papers that provide marmoset-based data here, as well.

We have also cited Sasaki et al. (2015), which studied synaptic development in marmoset (page 3, line 9).

8. Intro, “Moreover, the developmental expression pattern of some genes are primate-specific...” – IS primate specific

We thank the reviewer for pointing out the error. We have corrected this error (page 3, line 15).

9. Intro, “The common marmoset (*Callithrix jacchus*), a small New World monkey, is well suited to studies on neurodevelopmental diseases, and previous studies from our laboratory have described similarities in cortical architecture and synaptic development to humans (Oga et al., 2013; Sasaki et al., 2015)”. Both references, from the lab that also originates the present paper, deal with dendritic trees, not cortical architecture. This links to another problem that is brushed over in this paper, that is, the fact that the region from where the data were obtained is never documented, and there is little discussion of whether areas 8b and 9 in marmoset correspond to human areas.

One possibility is to cite here papers that have made the case that the areas being sampled are similar in architecture and connections in marmosets and humans (e.g. Burman et al. 2006 *J Comp Neurol* 2006 Mar 10;495(2):149-72; Reser et al. 2013 *Cereb Cortex* 23(8):1901-22) and then document the slices relative to the Paxinos atlas, as suggested above. Otherwise, the similarity should be documented perhaps in supplementary figures.

We thank the reviewer for pointing out the incorrect citing of literature. We have modified the phrase “cortical architecture” to “neuronal structure” (page 3, line 20). Further, we have made a figure that compares the slices with the Paxinos atlas (Suppl. Fig. 1).

10. Intro, “ages of birth (0M), 3 months (3M), and 6 months (6M), which can be taken to correspond to neonatal, infancy, and adolescence periods, respectively” – this is questionable. IN the Sawiak et al. paper (see above) it is proposed that marmoset adolescence is between 8 and 20 months (please note that sexual maturity is usually achieved by 12-13 months). Thus,

the present paper may in fact be dealing with pre-pubertal marmosets even at the oldest time point. The developmental time line proposed by Sawiak also seems to be more in line with the Burman et al. 2007 results (both papers deal with cortical development, and address areas 8 and 9).

As the reviewer pointed out, accurate aligning of ages between the model animal and humans is essential for this study, and we have carefully considered literature on the timeline of development in both marmoset and humans.

As the reviewer pointed out, Sawiak et al. (2018) stated that adolescence in marmoset is between 8-20 months, but also stated that puberty is 6-12 months; therefore, 6 months is at the beginning of puberty, and we have used the term “puberty” instead of “adolescence” throughout the revised manuscript (for the Abstract, page 2, lines 5 and 8). We also incorporated the data on the gray matter volume; the gray matter volume reaches a peak at 6 months in area 13 in marmoset (Sawiak et al. 2018), and 10-11 years in humans (Gogtay and Thompson 2020), which is also in puberty. This has been stated in the first paragraph of the Results section (page 5, line 14). We have also cited Burman et al. (2007) in the Discussion section, where we have discussed the limitation of this study (page 18, line 2).

11. Results, “Therefore, 3M in marmosets could correspond to ~3 years in humans” – this is close, but according to Sawiak 3M would correspond to a younger human. Not sure this makes a difference, but maybe allow for the possibility (e.g. 2-3 years).

We agree with the reviewer’s comment. We have modified the expression to 2-3 years (page 5, line 19).

12. Results, “cortical pyramidal neurons of Brodmann area 8b/9” – state layer (L3 presumably) and how the areas were identified.

We thank the reviewer for the comment. We have stated that we recorded from layer 3 (page 5, line 22). We identified layer 3 by microscopic observation (layer with relatively sparse pyramidal neurons), and confirmed by Nissl staining of some slices, which has been shown in Suppl. Fig. 1. We have also described how we identified layer 3 in the Methods section (page 21, line 10).

13. Results, vocalization: “the dorsomedial prefrontal cortex is among the brain regions

involved in vocalization in primates (Miller et al., 2010)”. This statement, as presented, is slightly misleading. The Miller paper does not identify areas 8 and 9 as being the places where c-fos labeled cells were found. Instead they state: “Based on descriptions from previous research (Brodman, 1909; Burman et al., 2006), the two most rostral regions in our analysis (Frontal Regions 1 and 2) showing high levels of cFos expression likely correspond to ventral-prefrontal cortex area 12/45, whereas the populations in the most caudal region (Frontal Region 3) are likely premotor cortex areas 6V and 6D. Future research will aim to refine this analysis and ascertain the functional role and anatomical boundaries of these neuronal populations”.

As the reviewer pointed out, Miller et al. (2010) did not identify areas 8 and 9 as responsive areas. Therefore, we have instead cited Loh et al. (2017), which showed that pre-supplementary and supplementary motor areas that receive projection from area 8 are activated (page 9, line 22).

14. Results, vocalization: “These findings are indicative of the existence of behavioral abnormality in VPA animals following abnormal synaptic development between birth and infancy”. The conclusion here amounts to a generic statement, to the effect that something changes in the vocalization, and something changes in the brain, so the two must be related. I don't think that the paper presents sufficient evidence to establish a causal link between the two sets of observations. This can be presented as another evidence of ASD-like symptoms, but try not to stretch too far.

We agree with the reviewer's comment that our data do not show a causal link between the synaptic phenotypes and behavioral abnormalities. An important purpose of our behavioral study was to demonstrate that behavioral abnormality is already present in the infantile marmoset as in human child, and we have clarified this point (page 9, line 18).

15. The discussion should mention whether marmosets and humans are born at corresponding developmental stages, or if not which species is more immature at birth. The paper reports differences between marmosets and humans at birth, and knowing the developmental stages would be important to interpret the significance of this finding. If I understand correctly the discussion on gene changes does allow for the possibility that OM marmoset corresponds to a younger human stage, but not for the possibility that marmosets are born more mature than humans.

As the reviewer pointed out, this is an important issue for our study. Although anatomical

evidence is limited, comparable elevation in the blood testosterone level in the perinatal period in both humans and marmoset (McKinnel et al. 2001; Forest et al. 1976) suggests a similar developmental stage at birth for both species. We have stated this in the Results section (page 5, line 17).

Reviewer #2 (Remarks to the Author):

This study by Watanabe and colleagues treats marmosets with valproic acid as a model of ASD. The authors then carry out a number of electrophysiological, neuroanatomical, behavioral, and genomic experiments. In general, the individual experiments are mostly carried out appropriately and results are described well. I found it intriguing that the marmoset genomic datasets overlapped better with the human ASD datasets than the rat valproic acid model, supporting the use of the marmoset.

We thank the reviewer for favorable comments. As the reviewer pointed out, we clarified better replication of human ASD by the marmoset model than by the rat VPA model, which is an important message of this study. In the revised manuscript, we have also compared the gene expression modulations in other rodent models with those in human ASD and obtained results that further support the utility of the marmoset model for human ASD. We have carefully revised the manuscript according to the reviewer's insightful comments, as listed below.

However, I think the manuscript suffers a bit in organization of how the data are integrated across the three time points and discussed.

Furthermore, these new results could be better discussed within the context of the previous literature.

We studied synaptic phenotypes and gene expression modulations in early postnatal development. Synapses develop extensively, and gene expression changes drastically in this period, and developmental dynamics of genes are related to molecular functions, as suggested by gene coexpression analysis (Parikshak et al. 2013). In this study, temporally related modulations in synaptic structure or function and gene expression were analyzed. Integrating these data, we revealed two different groups of phenotypes, as summarized in Fig. 6. In addition, behavioral analysis was carried out to verify ASD-like abnormality in infancy (previously, it was only studied in adult animals).

We have cited literature especially on human ASD (Hutsler and Zhang 2010; Tang et al. 2014) in the Discussion (page 13, line 22) and compared our results with the literature. We have

also cited recent papers that suggest distinct transcriptome or metabolome between rodents and humans (Bakken et al. 2016; Pembroke et al. 2021; Koopmans et al. 2018).

At the end of the paper, it is challenging to have a clear take away message with specific targets or time points that underlie the changes in synaptic physiology. This is mostly because results vary widely across the three time points depending on the experiments.

Some results are significant and others are not, depending on age and/or experiment. Perhaps the authors need to make a schematic chart that integrates all the data at each time point, or maybe even one or two of the time points should be eliminated.

We appreciate the suggestion by the reviewer. The different time courses of phenotypes were difficult to understand in the original manuscript, although this is the essential part of our study. We have made a schematic summary of the results (Fig. 6) and referred to this scheme where appropriate (for example, page 8, line 3, and page 10, line 10). We hope this will help the reader understand the two different patterns in the time course and their relationship with gene expression modulations.

In addition to this “tightening” of the story, I have some other minor suggestions:

1. The use of the specific time points should probably reference McKinnell et al., 2001.

As the reviewer suggested, we determined the specific time points based on the blood testosterone level (McKinnell et al. 2001) as well as other factors (synaptic density and cortical volume). We have described how we selected the specific time points and how they are compared to human development (page 5, line 17).

2. The authors mentioned in the animals' section that some animals in the UE group received 10% glucose (vehicle), which was used to make 4% VPA solution. Why did they give this vehicle to only some animals and not to all?

We thank the reviewer for pointing out the important issue about the animals. Ideally, we should have used only vehicle-treated animals as controls, but we also used untreated animals because of the limited availability of vehicle-treated animals. However, we confirmed that both treated and untreated groups showed similar development, since the body weight at birth was comparable between the two groups, as mentioned in the Methods section.

3. In few of their experiments, the number of samples for 0M and 6M are many fewer than the 3M group. Any specific reason for this?

The differences in the number of animals between the ages were also due to limited availability of animals. However, we believe that we have a sufficiently large number of data to detect phenotypes at each age.

4. I could not understand or find a good explanation for the synaptic changes in different parameters in one group and not in other.

We found two types of time courses in the synaptic phenotypes. The first type included dendritic spine density and miniature synaptic current frequencies, which were most affected in the early stage of synaptogenesis. The second type included LTD, spine size, and E/I balance, which were most affected in the infantile age. The grouping was validated by the existence of corresponding gene expression modulations. This has been summarized in the schematic (Fig. 6), which we hope will help the readers better understand the results.

5. The authors find changes in the synaptic structure with VPA treatment only in the 3M group. Did they t check in other groups as well?

In the original manuscript, we only presented data at 3M because the spine size was significantly different only at this age. However, as the reviewer pointed out, the data at other ages would also be informative. Therefore, we have examined synaptic structure at 6M and clarified that synaptic structure was not affected at 6M (Fig. 2e). We also wanted to clarify synaptic structure at 0M (where we expected to see no difference either), but unfortunately, we were unable to present the data because of the limited availability of samples.

6. The authors used DAPV in that 0M group to show that it blocks NMDA dependent LFS-induced LTD and say that probably NMDA dependent LTD is enhanced in the 3M group. However, they should do the same DAPV blockage in the 3M group and then show that after blockage with DAPV, there is no difference in the LTD between VPA and UE animals to confirm that the change in LTD is NMDA dependent.

We did not test NMDA dependence at 3M because of the limited availability of samples. As the reviewer pointed out, we have no direct evidence to tell that the LTD in VPA animals at 3M is NMDA-dependent. Therefore, we have modified the text and mentioned that it might be NMDA-dependent, but it was not tested (page 8, line 22).

7. In the introduction, the authors claim 85% of ASD is idiopathic with no genetic cause, but do not provide a citation to back that claim. I believe there are many genetic studies now showing that the percentage of known genetic causes is significantly higher than 15%.

As the reviewer pointed out, many ASD cases are linked with genetic causes, as high concordance rates between twins indicate (88% for monozygotic twins). Indeed, a large number of ASD-related genes (such as SFARI genes) have been reported. However, according to the latest literature (Casanova et al. 2020), genetic causes are not identified in 85% of ASD, which is categorized as idiopathic. This implies that ASD from single gene mutations is relatively rare, and multiple genetic causes underlie most of ASD. We also speculate that the VPA marmoset model well replicates the multi-gene expression modulations in human ASD. Accordingly, we have modified the text (page 2, line 18) and made clear that these cases have no identified link to genetic causes, although unidentified genetic causes are speculated to be highly prevalent (Myers et al. 2020), with citation of the literature.

8. In Fig 1g, at the 6 month timepoint, the log fold change of the 3 time points relative to one another is different but the authors do not comment on this.

The slightly different time courses of logFC in the spine density, mEPSC frequency, and mIPSC frequency were due to different effects of VPA exposure on these parameters. This is probably the cause of the distinct time course for E/I balance modulation (Fig. 1j). We have clarified the different time courses and implications for other synaptic phenotypes (page 7, line 18).

9. In Fig 2d, the authors do EM of synapses at 3M. Have the other 2 time points been examined?

We have also examined synaptic structure at 6M and added to Fig. 2e, as stated in the response to comment #5.

10. In the discussion, the authors use the term “mature” several times. But, I assume, they really

mean adolescent?

We thank the reviewer for pointing out the ambiguous use of the term “mature”. We have changed the term to “puberty” to reduce ambiguity (page 13, lines 16 and 19).

11. In the discussion, the authors mention LTD is required for vocal learning in songbirds. Since the 3M VPA monkeys have enhanced LTD, how do the authors reconcile this with the abnormal vocalizations?

As the reviewer pointed out, the enhanced LTD in the ASD model is not consistent with the requirement of LTD for vocal learning in songbirds. However, abnormal LTD is observed in various neurological states, including ASD. We have removed the sentence describing the requirement of LTD for vocal learning and cited a paper describing abnormal LTD in ASD model animals (page 15, line 7).

Reviewer #3 (Remarks to the Author):

This paper reports the results of studies in the valproic acid (VPA) model of autism in the marmoset. Three lines of research are described. 1) Electrophysiological studies of in vitro slices taken from the frontal cortex 2) vocalization studies of the living animals and 3) gene expression analyses of tissue taken from both frontal and temporal lobes. Most of this work was done in marmosets who were newborn, 3-months-old or 6-months-old.

We are grateful for the insightful comments, especially on the rationale of the study, as well as pointing out a number of important issues. We have carefully revised the manuscript according to the reviewer’s comments, and we feel that the reviewer’s comments helped us to significantly improve the paper.

Critique:

This is an ambitious series of studies based on the premise that the nonhuman primate is more appropriate for modeling autism than rodent models. A second premise is that the VPA model is a valid model for human autism.

As the reviewer pointed out, this study was initially based on these premises, as suggested by

comparative transcriptome and proteome analyses (Bakken et al. 2016; Pembroke et al. 2021; Koopmans et al. 2018) (page 3, line 15). Our transcriptome analysis indicated that a large number of gene modules show similar modulation between the marmoset VPA model and human ASD, and the similarity was stronger than between the rodent VPA model and human ASD. Therefore, VPA marmoset model replicates a broad range of transcription modulations in human ASD, and the premises have been substantiated.

The paper is not well written so it is difficult to follow the flow of arguments presented by the authors. It is curious that the rationale for conducting the three levels of investigation outlined above was not presented in this paper.

We thank the reviewer for pointing out that the rationale for the study was not well presented. In the early postnatal period, synapses develop extensively, and gene expression changes drastically. Developmental dynamics of genes are related to molecular functions, as revealed by gene coexpression analysis (Parikshak et al. 2013). In our study, temporally related modulations in synaptic structure or function and gene expression were combined to link ASD-related synaptic phenotypes with molecular functions. Using this approach, we elucidated molecular components that presumably underlie synaptic abnormalities. In addition, a behavioral analysis was carried out to verify ASD-like abnormality in infancy (previously, it was only studied in adult animals). We have stated the rationale for conducting these experiments more clearly in the introduction (page 4, lines 7 and 18).

Neither does the paper adequately make the case that the NHP provides more useful data than the many dozens of rodents VPA studies.

As the reviewer pointed out, the original manuscript did not mention much about the validity of the marmoset VPA model over rodent models. Therefore, we have compared the gene expression modulations in various ASD rodent models and human ASD based on coexpression modules. We found that a larger number of modules showed a similar modulation between the marmoset VPA model and human ASD than between rodent models and human ASD, suggesting that the marmoset model replicates human ASD more accurately than rodent models at the molecular level. We speculate that this is due to different developmental trajectories of gene expression between rodents and primates, which might confer different vulnerability of the genes to ASD-inducing factors. We have shown the results of the analyses in Suppl. Fig. 8 and described in the text (page 4, line 22; page 13, line 4; and page 16, line 21). We appreciate the reviewer's comment since it has helped us to clarify further evidence for the validity of the

marmoset VPA model for ASD research.

There are also many statements made throughout the paper that are not substantiated by the literature. For example, “Synaptic overdevelopment in the mature marmoset model was in parallel to human ASD.” This statement has no references to back it up and as far as I know, this “fact” has not yet been established using human tissue.”

We thank the reviewer for pointing out the absence of appropriate literature, especially on human ASD. There are studies in the postmortem human brain that showed an increased synaptic density in ASD (Hutsler and Zhang 2010, Tang et al. 2014). In the sentence pointed out above, as well as in other parts in the Introduction (page 6, line 16) and Discussion (pages 13, line 22), we have cited these papers.

I congratulate the authors on attempting a very complex series of studies. But, I would encourage them to recast this paper in a more focused manner. E/I balance, language development and gene expression do not necessarily fit in the same paper.

I would also encourage a more accurate representation of the autism literature.

Much of what is presented as well-established fact is based on initial or unreplicated studies.

A more circumspect approach to representing the state of autism research would be very welcome.

We thank the reviewer for the positive comment on this study. We agree with the reviewer that the manuscript was not written in a well-focused manner. We have carefully revised the manuscript and integrated the results of different experiments. We hope that the revised manuscript is more consistent and easier to understand, although we did not remove any data from the manuscript according to the Editor’s advice. We have also cited multiple papers on ASD pathology which were missing in the original manuscript, as described above. We are grateful for the reviewer’s comments, as they helped us to significantly improve the manuscript.

Reviewers' Comments:

Reviewer #1:

Remarks to the Author:

The authors have introduced appropriate changes in response to my initial comments. I address them according to the numbered points in the rebuttal letter.

1, 2: I appreciate the inclusion of more data about spine densities, which address one of my main comments.

3. I appreciate the inclusion of the additional references, but this has not been done to an appropriate extent that places this paper in appropriate context. The issue of the timeline of maturation of V1 versus frontal areas remains unresolved, and just mentioning this does not appear sufficient. The Bakken paper uses indirect measures to chart the development, whereas Huttenlocher uses direct measures of synaptogenesis. The hypothesis of late maturation of the frontal lobe is in agreement with other lines of evidence (e.g. cortex volume – Sawiak; cytoskeleton proteins – Burman). I believe that careful Discussion of this point needs to be included, since this is central to the interpretation of the origin of ASD-like symptoms, which are the central reason for the study. In other words, considering what is known about the time line of maturation of V1 versus frontal areas, were the present samples collected at times where they can shed light to the problem? What arguments can the authors provide in favour and against this possibility? If this is unclear, can this limitation be acknowledged, and new experiments be proposed?

4. See point above. Considering how late other studies propose that the maturation proceeds, would the authors be prepared to explicitly discuss this limitation, and propose future work that can resolve this issue? At present the discussion is cursory, being limited to a couple of sentences which don't even hint at the extent of this issue, i.e.: "Third, difference between areas was not addressed in this study. Within the cortex, some studies suggest different time courses of development between areas 7 and layers 63". (Note: in reality both studies demonstrate differences between areas, but 63 also shows between layers). For example, at the time points when the samples were obtaining, what was happening to the sampled layer in area 8b according to the previous studies? Was it largely mature, or still during a phase of rapid change?

5. Some elements of Figure S1 are essential to the paper, in particular the documentation of slice location relative to the Paxinos atlas. I suggest that this information needs to be incorporated to the main text. In my view, knowing where exactly the samples were obtained is essential, particularly considering the possibility of differences between areas.

6. The changes made in response to this point were satisfactory.

7. Citation of the Missler papers here would also be appropriate, given that these were the pioneering and still most comprehensive study of the subject.

8. OK

9. I appreciate the supplementary figure, but this alone is not sufficient, without further context. The simplest solution here would be to include citations, if the criteria used exactly corresponded to those proposed by an earlier study (e.g. after describing the characteristics of the area chosen for sampling, state "these criteria correspond to those described by...XXX...in the marmoset").

10. Puberty is indeed more appropriate than adolescence, so this is fine. As mentioned above, it may be appropriate to acknowledge explicitly in the discussion that the maturation of the frontal cortex extends far beyond the 6 month mark, both in terms of areas and layers within an area.

11. OK

12. This point is well addressed, although as mentioned above I believe that some elements of Figure S1 are essential to the paper and should be incorporated in a main figure.

13. I am not sure this change was helpful since it cannot be assumed that macaques and marmosets have the same connections. This argument should be based on papers that describe the connections of marmoset areas 8b and 9. Do these areas connect to the vocalisation-related areas of the marmoset frontal lobe, proposed by Miller (i.e. areas 12/45, 6D and 6V)?

14. The subtle changes here have helped address my comment, OK now.

15. OK

Reviewer #2:

Remarks to the Author:

The authors have addressed all of my concerns.

Reviewer #4:

Remarks to the Author:

The authors have addressed the critiques with minor specific revisions, which have made the overall response somewhat adequate, but two issues remain;

1) Assembly and integration of data sets and the flow of arguments. This is partly because of the amount of data packed without enough discussion and integration toward the core concept, which I believe is introduction of a valid primate model of ASD, supported by morphological, electrophysiological, behavioural and genomic data. While the 4 main components, are important, it would be much better to concentrate on the most relevant of these approaches/data sets, and leave the rest for a more specific paper.

The example that I can best comment upon is the electrophysiology data set. To make sense of changes in synapses in line with spine morphology and genomics (cluster 1 genes), the spontaneous EPSP/IPSP recordings are adequate (Fig 1 & Fig 6). However, addition of LTD experiments with a view to support synaptic plasticity changes is premature, so as critical period plasticity and E/I ratio. The involvement of LTD as a form of synaptic plasticity in critical period regulation remains a matter of debate in neuroscience, and thus it can't be directly related to Cluster 2 genes as shown in Fig 6. It would be much better to cite references for the relevance of cluster 2 and 3 genes to plasticity, either synaptic or during the critical period, rather than attempting to get some data for the purpose. I believe vocalisation changes are also a behavioural representation of another critical period plasticity which has not been incorporated/ discussed well in relation to genomics in Fig 6.

For the above reasons, the present title does not best represent the data which are being presented.

2) To adequately make the case that the NHP provides more useful data than the many dozens of rodent VPA studies, it must be clearly mentioned in the abstract or as concluding remarks that the greater similarity of transcriptome analysis/gene modules between the marmoset VPA model and human ASD, revealed in this study, places the marmoset model as more appropriate model than rodent models. This is mainly because none of the other studied aspects makes a direct comparison between findings in marmoset and the previous data in rodent.

Point-by-point response to the reviewers' comments

Reviewers' comments are shown as blue text. Changes to the manuscript are highlighted in yellow.

Reviewer #1 (Remarks to the Author):

The authors have introduced appropriate changes in response to my initial comments. I address them according to the numbered points in the rebuttal letter.

We are grateful for the efforts by the reviewer to reconsider our manuscript. We appreciate the valuable comments that raised important issues that need to be addressed. We have carefully read the comments and revised the manuscript accordingly.

1, 2: I appreciate the inclusion of more data about spine densities, which address one of my main comments.

3. I appreciate the inclusion of the additional references, but this has not been done to an appropriate extent that places this paper in appropriate context. The issue of the timeline of maturation of V1 versus frontal areas remains unresolved, and just mentioning this does not appear sufficient. The Bakken paper uses indirect measures to chart the development, whereas Huttenlocher uses direct measures of synaptogenesis. The hypothesis of late maturation of the frontal lobe is in agreement with other lines of evidence (e.g. cortex volume – Sawiak; cytoskeleton proteins – Burman). I believe that careful Discussion of this point needs to be included, since this is central to the interpretation of the origin of ASD-like symptoms, which are the central reason for the study. In other words, considering what is known about the time line of maturation of V1 versus frontal areas, were the present samples collected at times where they can shed light to the problem? What arguments can the authors provide in favour and against this possibility? If this is unclear, can this limitation be acknowledged, and new experiments be proposed?

We thank the reviewer for suggesting to discuss the implications of neural maturation for ASD symptoms. We have added a new paragraph at the beginning of the Discussion (page 13, line 21) to address this issue.

In the present study, we aligned the model marmoset to human ages, compared the phenotypes in marmoset with those in humans, and predicted early age pathology of human ASD. Comparing the peak spine density in the PFC, the age of 3 months of marmoset seems to correspond to about 2-3 years of age (Huttenlocher et al. [1997], Petanjek et al. [2011]).

Symptoms of ASD become apparent and typically diagnosed around this age. Consistently, we revealed an ASD-like behavior in marmoset at 3 months. It has been hypothesized that symptoms at infancy lead to development of pathology in later ages. As expected, we observed an increase in spine density in the model marmoset at 6 months, which was also a phenotype of human ASD in puberty (Tang et al. [2014]). The replication of age-dependent human pathology in the model marmoset implies that these time points are appropriate for the study of ASD pathology.

We have further considered differences in neuronal development between areas in the marmoset. As the reviewer pointed out, neurons in the prefrontal cortex (PFC) develop more slowly than those in V1, resulting in a caudo-rostral gradient of development. Although the peak times of spine density between the PFC and V1 were similar (3 months; Oga et al. [2013], Sasaki et al. [2015], Missler et al. [1993]), PFC neurons showed a slower decline than in V1 (Oga et al. [2013], Sasaki et al. [2015]). The cortical volume (Sawiak et al. [2018]) also shows similar peak times (6 months for PFC, 5 months for V1) but later onset of decline for PFC (11 months for PFC, 6 months for V1). The gray matter density is an indirect measure of a complex architecture of glia, vasculature, and neurons with dendritic and synaptic processes. The developmental changes in the non-phosphorylated neurofilament protein (Burman et al. [2007]) also suggest that layer 3 neurons in the PFC are not fully mature at 6 months. We have discussed that the difference in neuronal maturation between puberty and adult ages suggests different phenotypes in adults from those we revealed in the present study, and that adult phenotypes need to be clarified in future studies. We have also discussed the potential need for different therapeutic approaches for young and adult people with ASD.

4. See point above. Considering how late other studies propose that the maturation proceeds, would the authors be prepared to explicitly discuss this limitation, and propose future work that can resolve this issue? At present the discussion is cursory, being limited to a couple of sentences which don't even hint at the extent of this issue, i.e.: "Third, difference between areas was not addressed in this study. Within the cortex, some studies suggest different time courses of development between areas 7 and layers 63". (Note: in reality both studies demonstrate differences between areas, but 63 also shows between layers). For example, at the time points when the samples were obtaining, what was happening to the sampled layer in area 8b according to the previous studies? Was it largely mature, or still during a phase of rapid change?

As we have stated in the point above, we agree with the reviewer that layer 3 neurons in area 8b/9 are not fully mature at the latest time point we studied. The delayed maturation suggests that phenotypes in adulthood may be different from those we observed in the present study.

We need to clarify phenotypes at adult ages in future studies because most of the work on human ASD pathology has been done at adult ages. We have discussed this point in the Discussion (page 14, line 14).

5. Some elements of Figure S1 are essential to the paper, in particular the documentation of slice location relative to the Paxinos atlas. I suggest that this information needs to be incorporated to the main text. In my view, knowing where exactly the samples were obtained is essential, particularly considering the possibility of differences between areas.

As suggested by the reviewer, we have moved the documentation of slice location (photographs of the slices and drawing) from supplementary Figure 1 to main Figure 1.

6. The changes made in response to this point were satisfactory.

7. Citation of the Missler papers here would also be appropriate, given that these were the pioneering and still most comprehensive study of the subject.

We agree with the reviewer that the Missler paper is a highly important work on neural development in marmoset. Therefore, we have cited this paper in the Introduction (page 3, line 9).

8. OK

9. I appreciate the supplementary figure, but this alone is not sufficient, without further context. The simplest solution here would be to include citations, if the criteria used exactly corresponded to those proposed by an earlier study (e.g. after describing the characteristics of the area chosen for sampling, state “these criteria correspond to those described by...XXX...in the marmoset”).

We have described the characteristics of the cytoarchitecture in our samples and compared it to literature (Burman et al. J. Comp. Neurol. [2006]), which described regional difference of the cytoarchitecture in the frontal cortex of marmoset. We have stated how we confirmed the areas in the Methods section (page 21, line 9).

10. Puberty is indeed more appropriate than adolescence, so this is fine. As mentioned above, it may be appropriate to acknowledge explicitly in the discussion that the maturation of the

frontal cortex extends far beyond the 6 month mark, both in terms of areas and layers within an area.

We agree with the reviewer that layer 3 neurons in the prefrontal cortex develop relatively slowly and are not fully mature at 6 months. We have explicitly discussed this point in the Discussion (page 14, line 14).

11. OK

12. This point is well addressed, although as mentioned above I believe that some elements of Figure S1 are essential to the paper and should be incorporated in a main figure.

As suggested by the reviewer, we have moved the photographs and drawing of the slices from supplementary Figure 1 to main Figure 1.

13. I am not sure this change was helpful since it cannot be assumed that macaques and marmosets have the same connections. This argument should be based on papers that describe the connections of marmoset areas 8b and 9. Do these areas connect to the vocalisation-related areas of the marmoset frontal lobe, proposed by Miller (i.e. areas 12/45, 6D and 6V)?

We have cited Majka et al. Nat. Commun. (2020), which revealed connections between areas 8b/9 and 45/6D/6V in marmoset (page 9, line 22).

14. The subtle changes here have helped address my comment, OK now.

15. OK

Reviewer #2 (Remarks to the Author):

The authors have addressed all of my concerns.

Reviewer #4 (Remarks to the Author):

We are grateful for the insightful comments on the manuscript. We have carefully revised the manuscript to better present the rationale of the study and improve the flow of arguments. The

reviewer's comments helped us to improve the manuscript significantly.

The authors have addressed the critiques with minor specific revisions, which have made the overall response somewhat adequate, but two issues remain;

1) Assembly and integration of data sets and the flow of arguments. This is partly because of the amount of data packed without enough discussion and integration toward the core concept, which I believe is introduction of a valid primate model of ASD, supported by morphological, electrophysiological, behavioural and genomic data. While the 4 main components, are important, it would be much better to concentrate on the most relevant of these approaches/data sets, and leave the rest for a more specific paper.

As the reviewer pointed out, the core concept of this paper is introduction of a valid primate model of ASD. To evaluate the validity of the model, both behavioral phenotypes (vocalization) and endophenotypes (synapses and gene expression) should be identified. Therefore, we believe that all these data sets are essential for the introduction of this primate model. That said, we also agree with the reviewer that there was still a need for improvement concerning the integration of the data sets. Therefore, we have revised the Discussion section and clarified how each data is related to ASD pathology and to each other, as stated below.

The example that I can best comment upon is the electrophysiology data set. To make sense of changes in synapses in line with spine morphology and genomics (cluster 1 genes), the spontaneous EPSP/IPSP recordings are adequate (Fig 1 & Fig 6).

However, addition of LTD experiments with a view to support synaptic plasticity changes is premature, so as critical period plasticity and E/I ratio. The involvement of LTD as a form of synaptic plasticity in critical period regulation remains a matter of debate in neuroscience, and thus it can't be directly related to Cluster 2 genes as shown in Fig 6. It would be much better to cite references for the relevance of cluster 2 and 3 genes to plasticity, either synaptic or during the critical period, rather than attempting to get some data for the purpose. I believe vocalisation changes are also a behavioural representation of another critical period plasticity which has not been incorporated/ discussed well in relation to genomics in Fig 6.

As the reviewer commented, the development of LTD is not in parallel with the emergence of the critical period (Jiang et al. J. Neurosci. 2007), and its role in critical period regulation has been debated. Therefore, we have revised the discussion on LTD (page 15, line 8) and avoided speculative discussions on the relationship between LTD and critical periods. We have focused on critical period-related genes, which support the critical period hypothesis of ASD

(page 15, line 13). We have also revised the discussion on the relationship between vocalization development and synaptic development (page 15, line 23). Particularly, one of the critical period-related genes is related to the development of vocalization in rodents (page 16, line 1), which suggests a relationship between vocalization changes and critical period plasticity. To clarify the relationship with synaptic and molecular phenotypes, we have added vocalization modulation (reduction in entropy) to Figure 6.

For the above reasons, the present title does not best represent the data which are being presented.

We thank the reviewer for the helpful suggestion. We have modified “circuit remodeling” to “plasticity”, because the latter represents the results more directly. Since the validity of the marmoset model for human ASD is an important message of this study, we have also added the phrase “with high molecular fidelity”.

2) To adequately make the case that the NHP provides more useful data than the many dozens of rodent VPA studies, it must be clearly mentioned in the abstract or as concluding remarks that the greater similarity of transcriptome analysis/gene modules between the marmoset VPA model and human ASD, revealed in this study, places the marmoset model as more appropriate model than rodent models. This is mainly because none of the other studied aspects makes a direct comparison between findings in marmoset and the previous data in rodent.

We thank the reviewer for pointing out that the similarity of transcriptome between the marmoset model and human ASD was not clearly mentioned. In the Discussion, we have described details about gene modules that were commonly modulated in the marmoset model and human ASD but not between the rodent models and human ASD (page 16, line 5). Among these were glial cell-related modules, and glial cells have essential roles in the regulation of synapses. In the last paragraph of the Discussion, we have stated that the higher number of concordant genes between the marmoset model and human ASD indicates that the marmoset model is more appropriate than rodent models (page 17, line 11).

Reviewers' Comments:

Reviewer #1:

Remarks to the Author:

I congratulate the authors on the substantial improvement. My suggestions have been acted upon, so I only have some minor corrections to the reference list. I am happy to indicate that I have no other comments.

Point 13: this is fine, but there is something strange about the reference (43) itself - no page or paper numbers, title seems like a place holder?

Also on references:

33, 66-68. species name should be in italics

49. also seems incomplete - is this still in press?

Reviewer #4:

Remarks to the Author:

Authors have chosen to keep all the data sets and their revisions are limited to a few sentences in the discussion and revision of title. No major rearrangement has been made to help significantly change assembly and integration of data sets for a better flow of arguments, which could be partly due to the later arrival of my comments in the process and their settling with other reviewers' comments.

The manuscript however has benefited from these limited revisions made in response to my comments and overall, I am in favor of this publication and have no further comments.

Point-by-point response to the reviewers' comments

Reviewers' comments are shown as blue text. Changes to the manuscript are highlighted in yellow.

Reviewer #1 (Remarks to the Author):

I congratulate the authors on the substantial improvement. My suggestions have been acted upon, so I only have some minor corrections to the reference list. I am happy to indicate that I have no other comments.

Point 13: this is fine, but there is something strange about the reference (43) itself - no page or paper numbers, title seems like a place holder?

Also on references:

33, 66-68. species name should be in italics

49. also seems incomplete - is this still in press?

We thank the reviewer for considering the revised manuscript. We have corrected the information of references 43 and 49, and corrected the species name in references 33, 66, 67, and 68.

Reviewer #4 (Remarks to the Author):

Authors have chosen to keep all the data sets and their revisions are limited to a few sentences in the discussion and revision of title. No major rearrangement has been made to help significantly change assembly and integration of data sets for a better flow of arguments, which could be partly due to the later arrival of my comments in the process and their settling with other reviewers' comments.

The manuscript however has benefited from these limited revisions made in response to my comments and overall, I am in favor of this publication and have no further comments.

We thank the reviewer for assessing our revised manuscript and for the positive response.